

# Reassessment of the radiocesium resuspension flux from contaminated ground surfaces in East Japan

Mizuo Kajino[1,2,3], Akira Watanabe[4,5], Masahide Ishizuka[6], Kazuyuki Kita[7], Yuji Zaizen[1], Takeshi Kinase[8,1], Rikuya Hirai[9,10], Kakeru Konnai[9], Akane Saya[1], Kazuki Iwaoka[11], Yoshitaka Shiroma[12], Hidenao Hasegawa[13], Naofumi Akata[2], Masahiro Hosoda[2], Shinji Tokonami[2], and Yasuhito Igarashi[14,7]

[1]Meteorological Research Institute (MRI), Japan Meteorological Agency (JMA), Tsukuba, Ibaraki 305–0052, Japan
[2]Institute of Radiation Emergency Medicine (IREM), Hirosaki University, Hirosaki, Aomori 036–8564, Japan
[3]Faculty of Life and Environmental Sciences, University of Tsukuba, Tsukuba, Ibaraki 305–8572, Japan
[4]Faculty of Symbiotic Systems Science, Fukushima University, Fukushima, Fukushima 960–1296, Japan
[5]Institute for Climate Change, Fukushima, Fukushima 960–0231, Japan
[6]Faculty of Engineering and Design, Kagawa University, Takamatsu, Kagawa 761–0396, Japan
[7]Graduate School of Science and Engineering, Ibaraki University, Mito, Ibaraki 310–8512, Japan
[8]Institute of Arctic Climate and Environmental Research (IACE), Japan Agency for Marine-Earth Science and Technology (JAMSTEC) Yokohama, Kanagawa 236–0001, Japan
[9]School of Life and Environmental Sciences, University of Tsukuba, Tsukuba, Ibaraki 305–8572, Japan
[10]Research Center for Advance Science and Technology (RCAST), University of Tokyo, Meguro, Tokyo 153–0041, Japan
[11]National Institute of Radiological Sciences, National Institues for Quantum and Radiological Science and Technology, Chiba, Chiba 263–8555, Japan
[12]Faculty of Education, University of the Ryukyus, Nishihara, Okinawa 903–0213, Japan
[13]Institute for Environmental Sciences, Rokkasho, Aomori 039–3212, Japan
[14]Institute for Integrated Radiation and Nuclear Science (KURNS), Kyoto University, Kumatori, Osaka 590–0494, Japan

*Correspondence to*: M. Kajino (kajino@mri-jma.go.jp)





**Abstract.** Resuspension of $^{137}$Cs from the contaminated ground surface to the atmosphere is essential for understanding the environmental behaviors of $^{137}$Cs and estimating external and inhalation exposure of residents. Kajino et al. (2016) assessed the $^{137}$Cs resuspension flux from bare soil and forest ecosystems in East Japan in 2013 using a numerical simulation constrained by surface air concentration measurements. However, the simulation was found to underestimate the observed deposition amounts by two orders of magnitude. The reason for this underestimation is that the simulation assumed that resuspended $^{137}$Cs

is carried by submicron aerosols, which have low deposition rates. Based on the observational indications that soil dust and bioaerosols are the major carriers of resuspended $^{137}$Cs, a new simulation is performed with higher deposition rates constrained by both surface concentrations and deposition amounts. In the new estimation, the areal total annual resuspension of $^{137}$Cs in 2013 is 25.7 TBq, which is equivalent to 0.96% of the initial deposition (2.68 PBq). Due to the rapid deposition rates, the annual redeposition amount is also large at 10.6 TBq, approximately 40% of the resuspended $^{137}$Cs. The resuspension rate

through the atmosphere (0.96% y$^{-1}$) seems slow, but it (2.6×10$^{-5}$ d$^{-1}$) may not be negligibly small compared to the actual decreasing trend of the ambient gamma dose rate obtained in Fukushima Prefecture after the radioactive decay of $^{137}$Cs plus $^{134}$Cs in 2013 is subtracted (1.0–7.9×10$^{-4}$ d$^{-1}$): Resuspension can account for 1–10% of the decreasing rate due to decontamination and natural decay through land surface processes. The current simulation underestimated the $^{137}$Cs deposition in Fukushima city in winter by more than an order of magnitude, indicating the presence of additional resuspension sources.

The site of Fukushima city is surrounded by major roads. Heavy traffic on wet and muddy roads after snow removal operations could generate superlarge (approximately 100 µm in diameter) road dust or road salt particles, which is not included in the model but might contribute to the observed $^{137}$Cs at the site.

*Keywords:* Fukushima nuclear accident, resuspension of $^{137}$Cs, submicron and supermicron aerosols, dust and forest aerosols,

source-receptor analysis, seasonal budget.





## 1 Introduction

More than ten years have passed since the Fukushima Daiichi Nuclear Power Plant (F1NPP) accident. Extensive studies have been performed thus far using field observations, laboratory experiments, and numerical simulations aiming at a full
understanding of atmospheric dispersion and deposition of directly emitted radionuclides associated with the accident, which occurred in March 2011 (i.e., primary emission). It is difficult to cite all relevant papers here, so one can refer to review papers such as Mathieu et al. (2018), but a few remarkable studies are introduced here with some updates. Aircraft monitoring studies (NRA, 2012; Torii et al., 2012; 2013; Sanada et al., 2014) have provided the spatial distributions of radio-Cs and radio-I that were deposited to the ground surface in March 2011 over all of Japan. Tsuruta et al. (2014) and Oura et al. (2015) measured
the hourly surface air activity concentrations of $^{137}$Cs at 99 stations in East Japan. These two powerful spatiotemporal measurement datasets together with comprehensive emission scenarios provided by the Japan Atomic Energy Agency (e.g., Katata et al., 2015; Terada et al., 2020) enable us to identify transport and deposition events over the land surface in Japan (e.g., Tsuruta et al., 2014; Nakajima et al., 2017; Sekiyama and Iwasaki, 2018). These data were also useful to validate the numerical simulation results provided by various regional-scale atmospheric models (Draxler et al., 2015; Leadbetter et al.,
2015; Kitayama et al., 2018; Sato et al., 2018; 2020; Kajino et al., 2019; Goto et al., 2020) and were applied for other advanced numerical techniques, such as inverse modeling (Yumimoto et al., 2016; Li et al., 2019), ensemble forecasting (Sekiyama et al., 2021), and data assimilation (Sekiyama and Kajino, 2020).

In addition to spatial observations, detailed measurements have been helpful to investigate the mechanisms of atmospheric deposition and emissions from reactors. Kaneyasu et al. (2012) used size distribution measurements of multiple
chemical components obtained in April and May 2011 to indicate that submicron sulfate aerosols can be a major carrier of radio-Cs, and in fact, numerical simulations assuming hydrophilic submicron carrier aerosols have been successful (all models mentioned above made this assumption). On the other hand, Adachi et al. (2013) isolated hydrophobic supermicron Cs-bearing particles (referred to as Cs-bearing microparticles; CsMPs) from aerosol filters collected in March 2011; the atmospheric behaviors of these CsMPs could be quite different from those of hydrophilic submicron particles. Detailed analyses of CsMPs
are helpful for understanding emission events and mechanisms (Igarashi et al., 2019a; Kajino et al., 2021) and deposition processes (Dépée et al., 2019). Vertical measurements obtained on mountains (Hososhima and Kaneyasu, 2014; Sanada et al., 2018) have revealed the importance of cloud deposition over mountainous forests in East Japan. Even though the cloud deposition process is not included in other models, its importance has been inferred from some numerical simulations (Katata et al., 2015; Kajino et al., 2019).

A great number of numerical studies have been conducted for primary emissions, but only one numerical study (Kajino et al., 2016, hereinafter K16) has been performed on the atmospheric dispersion and deposition of radionuclides that have been resuspended from contaminated ground surfaces (secondary emissions). For primary emissions, the emission point is known, and many emission events can be identified, whereas for secondary emissions, the emission mechanisms are





unknown, and the ground surfaces (as emission sources) are highly heterogeneous. It is impossible to measure radio-Cs
resuspension fluxes from every ground surface, but knowledge has been accumulated from long-term atmospheric
measurements recorded at several locations. Ochiai et al. (2016) showed that the surface concentrations of $^{137}$Cs were high in
summer and low in winter in the contaminated forest area in the Abukuma Highlands. Ochiai et al. (2016) also showed that
the temporal variations in fine-mode (< 1.1 µm in diameter) and coarse-mode (> 1.1 µm) $^{137}$Cs behaved differently by season,
indicating that the major emission sources could be different between winter and summer. Nakagawa et al. (2018) conducted
size-resolved *n*-alkane and $^{137}$Cs measurements in similar forest areas and concluded that among biogenic emission sources,
epicuticular wax is less likely and bioaerosols such as pollen and fungal spores are more likely. Based on long-term
measurements taken in the same forest area, Kinase et al. (2018) indicated the association of mineral dust in late spring and
bioaerosols in summer and autumn. Kinase et al. (2018) also found that the contribution of the forest fires that occurred in
March 2013 to the surface $^{137}$Cs concentrations was negligibly small. Kinase et al. (2018) reported that the surface
concentration of $^{137}$Cs was positively correlated with the surface wind speed in winter but not in summer. Igarashi et al. (2019b)
further investigated the possible sources of $^{137}$Cs-rich bioaerosols in summer and suggested the substantial involvement of
fungal spores. Atmospheric humidity plays a key role in the discharge of fungal spores, which is consistent with the findings
of Kita et al. (2020), who stated that the surface concentration of $^{137}$Cs in mountainous forests became higher in the presence
of precipitation in summer. Cedar pollen particles could contain a considerable amount of $^{137}$Cs in the forest areas of Abukuma
Highlands, but they are emitted from late February to early May, not during summer (Igarashi et al., 2019b). In fact, number
of pollen particles was 1/10 of number of bacteria (including spores) or less in summer (Kinase et al., 2018; Igarashi, 2021).

The numerical simulations conducted by K16 were consistent with the findings described above: the surface
concentrations in the mountainous forest area are low in winter and high in summer, the contributions of mineral dust are high
in winter, and those of bioaerosols are high in summer. However, Watanabe et al. (2021) found that the simulations of K16
underestimated the observed deposition amounts by approximately two orders of magnitude. The major reason for this large
discrepancy in deposition is the incorrect assumption of the physical properties of resuspended $^{137}$Cs by K16. K16 constrained
the deposition efficiency of $^{137}$Cs in their simulations to be consistent with the primary emission period (March 2011), which
involved submicron carriers; however, based on the above-described measurements, the major carriers of $^{137}$Cs should be much
larger. In the current study, the deposition efficiency of $^{137}$Cs in the simulation is constrained to be consistent with the measured
concentration and deposition amounts in the resuspension period (i.e., 2013). The regional budgets of $^{137}$Cs are thoroughly
reassessed using more realistic model configurations, and the differences between the old and current estimates are clearly
compared in this study.





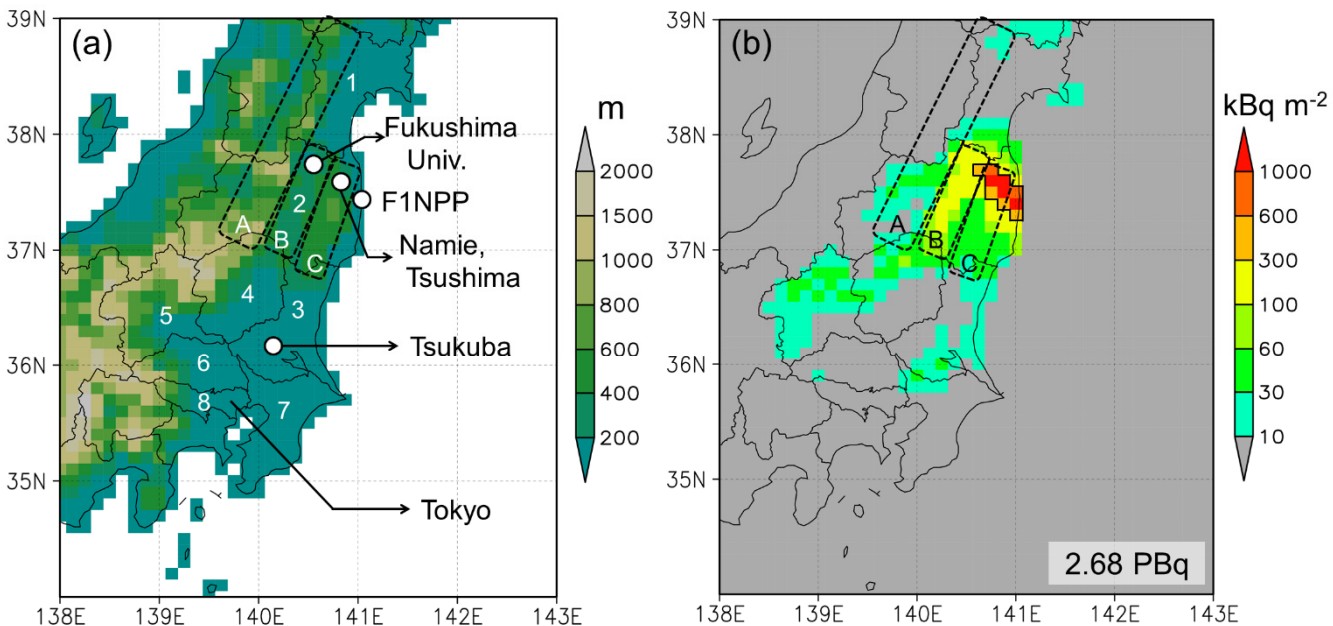

Name of prefectures: 1. Miyagi, 2. Fukushima, 3. Ibaraki, 4. Tochigi, 5. Gunma, 6. Saitama, 7. Chiba, 8. Tokyo
Geographical features: A. Ou Mountains, B. Nakadori Valley, C: Abukuma Highlands

**Figure 1:** (a) Model domain (Δ longitude = 0.125° and Δ latitude = 0.1°), terrestrial elevations, major locations, and names of prefectures and geographical features. (b) Initial deposition amounts of $^{137}$Cs measured by aircraft (NRA, 2012) and used as boundary conditions for the simulation. The decay correction for the observation was made for March-May 2012, depending on the regions. The area surrounded by the black solid line is defined as the resuspension source area (> 300 kBq m$^{-2}$) in the source-receptor analysis presented in Sect. 3.5. The total areal amount is embedded at the bottom right of the panel.

## 2 Methods

### 2.1 Observation data

To constrain the deposition rates and resuspension fluxes used in the simulations, activity measurement data containing surface concentrations and deposition amounts at three observation sites, one in a contaminated forest (Namie, Tsushima), one in an urban/rural area near the contaminated forest (Fukushima), and one in a downwind location (Tsukuba), are used (Fig. 1).

The Namie (Tsuhima) site is approximately 30 km northwest of the F1NPP, is located in the difficult-to-return zone (DRZ; >50 mSv y$^{-1}$), and is surrounded by forests in the Abukuma Highlands. The center of Namie town is located near the coast of the Pacific Ocean, but the observation site is surrounded by mountainous forests. Thus, to avoid confusion, the site is denoted as Namie (Tsushima) throughout the manuscript. The initial deposition amount indicated by the airborne measurements was 2300 kBq m$^{-2}$ (Fig. 1b), and decontamination work was not conducted in 2013. The locations at which the surface concentration measurements and deposition measurements are taken are different but are very close to each other (the





direct distance is approximately 400 m). The concentration measurements are conducted in the schoolyard of a high school (140.768°E, 37.562°N) (Ishizuka et al., 2017; Kinase et al., 2018), and the deposition measurements are made by Fukushima Prefecture at Tsushima Screening Center (140.765°E, 37.561°N) (data available at https://www.pref.fukushima.lg.jp/site/portal/genan225.html, last accessed June 30, 2021). The sampling intervals of the concentration and deposition measurements at this site are 1–2 days and 1 month, respectively.

The Fukushima site is in Fukushima city, located approximately 60 km northwest of the F1NPP. The Fukushima site is located in the Fukushima Basin in the Nakadori Valley, surrounded by the Ou Mountains (the peaks of which are 1000 – 2000 m in elevation) to the west and the Abukuma Highlands (the peaks of which are mostly lower than 1000 m in elevation) to the east (Fig. 1). The concentration and deposition measurements are conducted at Fukushima University (140.45°E, 37.68°N) (Watanabe et al., 2021). Fukushima University is located on a small hill at the southern edge of the Fukushima Basin and is surrounded by major roads. The distances from Route 4 and national highway E4 to the university are shorter than 1 km. The land use type of the site is characterized as urban/rural. The initial deposition amount indicated by the airborne measurement was 190 kBq m$^{-2}$ (Fig. 1b), one order of magnitude smaller than that at the Namie (Tsushima) site. Decontamination was conducted in 2013 in Fukushima city, and almost 90% of decontamination was completed for agricultural fields and public facilities by March 2014 (Watanabe et al., 2021). The achievement ratios of decontamination for other land use types are 50%, 9%, and 5% for residential areas, roads, and forests (only living areas; the removal of shrub and litter layers within 20 meters from the forest edges), respectively (Watanabe et al., 2021). The sampling intervals of the concentration and deposition measurements at this site are 3 days and 1 month, respectively.

The Tsukuba site is located in Tsukuba city, Ibaraki Prefecture, approximately 170 km southwest of the F1NPP. It is located in the eastern part of the Kanto Plain, the most populated area in Japan. The concentrations and deposition amounts are measured at the Meteorological Research Institute (140.13°E, 36.06°N) (Igarashi et al., 2015). The initial deposition amount was 21 kBq m$^{-2}$ (Fig. 1b), one order of magnitude smaller than that at the Fukushima site and two orders of magnitude smaller than that at the Namie (Tsushima) site. Decontamination was not conducted in most of the areas around this site due to the low ambient gamma dose rates. The sampling intervals of the concentration and deposition measurements at this site are 1 week and 1 month, respectively.

The locations, geographical features, and airborne-measured initial deposition amounts of these sites are visualized using Google Earth in the Supplement 1.

## 2.2 Numerical simulations

The Lagrangian model (LM) developed by K16 was used in this study. Thus far, a cumulus convection parameterization of Emanuel and Zivkovic-Rothman (1999) has been implemented to the model. The model description and simulation setup used in this study are identical to those of K16, except the cumulus convection parameterization, but are briefly repeated in this



section. The differences of simulations with and without the cumulus convection are presented later in Fig. 10 in Sect. 3.4. Errors between the original 1-D Eulerian model and the 1-D Lagrangian model developed and implemented to LM in the current study are summarized in Supplement 2. The LM considers the advection, turbulent diffusion, dry deposition, and wet deposition of atmospheric constituents. In the case of radionuclides, radioactive decay is also considered. As shown in Fig. 1a, the model domain covers the eastern part of Japan, from 138–140°E and from 34–39°N, with the same horizontal resolutions ($\Delta x$ is approximately 11 km; $\Delta$ longitude = 0.125° and $\Delta$ latitude = 0.1°) as the meteorological analysis data, the Grid Point Value-Mesoscale Model (GPV-MSM) of the Japan Meteorological Agency (JMA). The GPV-MSM provides data on three hourly meteorological variables on the surface and at vertical layers from 1000 hPa to 100 hPa. No meteorological models are applied to simulate finer-scale phenomena or to obtain detailed meteorological variables such as turbulent diffusivities or hydrometeor concentrations. The fundamental variables such as the wind field, temperature, humidity, and geopotential height obtained from the meteorological analysis data are interpolated horizontally and temporally and applied to simulate the locations and masses (or radioactivities) of Lagrangian particles. The simulation period is from December 1, 2012, to January 1, 2013, and the analysis period is the full year of 2013, from January 1, 2013, to January 1, 2014.

*2.2.1 Deposition schemes*

The key parameters used in this study are introduced below using equations. The LM does not include comprehensive deposition schemes; deposition processes are simply parameterized. The wet scavenging rate $\Lambda_{\text{wet}}$ ($s^{-1}$) is expressed as a function of the surface precipitation rate $P$ (mm $s^{-1}$) as follows:

$$\Lambda_{\text{wet}} = \frac{3}{4} \frac{E_c(a_m, r_m)}{a_m} P, \tag{1}$$

where $E_c$ is the collection efficiency of aerosols by the hydrometeor and $a_m$ and $r_m$ are the mean radii of the hydrometeor and aerosols, respectively. Empirically, $a_m$ is characterized by $P$ as $a_m = 0.35\, P^{0.25}$. $E_c$ is a function of $a_m$ and $r_m$, but, practically, a single constant value is used for each simulation. Eq. 1 is applied for all types of wet deposition. The differences among rain, snow, and graupel precipitation and the differences between in-cloud and below-cloud scavenging are not considered.

The dry scavenging rate $\Lambda_{\text{wet}}$ ($s^{-1}$) is expressed as follows:

$$\Lambda_{\text{dry}} = \frac{2}{z_{srf}} \left( 1 - \frac{z}{z_{srf}} \right) v_d, \tag{2}$$

where $z$ is the height of Lagrangian particles, $z_{srf}$ is the surface layer height set as 100 m in this study, and $v_d$ is the dry deposition velocity. $v_d$ depends on aerosol sizes and surface conditions such as wind speed, roughness, and land use types, but a single





constant value is applied in this study. We only consider the difference in $v_d$ over the land and the ocean; $v_d$ over the ocean is 0.1 times smaller than that over the land (K16).

Fog or cloud deposition plays a key role in the deposition of $^{137}$Cs over the mountains in East Japan (Hososhima and Kaneyasu, 2015; Katata et al., 2015; Sanada et al., 2018; Kajino et al., 2019; Imamura et al., 2020) but is not considered in the
study because the GPV-MSM product does not provide fog data (or cloud water in the bottom layers of the model grids).

In K16, we determined an $E_c$ value of 0.04 and a $v_d$ over land (simply referred to as $v_d$ hereinafter) value of 0.1 cm s$^{-1}$, so the initial deposition amount of $^{137}$Cs over land (2.53 PBq) simulated using the emission scenario of Katata et al. (2015) was closest to that observed (2.68 PBq, see Fig. 1b) among the various sensitivity simulations. However, the major carriers of $^{137}$Cs during primary emissions (i.e., the direct emissions associated with the nuclear accident) are submicron particles (several
100 nm in diameter, e.g., Kaneyasu et al., 2012); thus, the optimized deposition parameters are the orders of these submicron particles. However, the major carriers of $^{137}$Cs resuspended from the ground surface could be supermicron particles such as soil dust and bioaerosols (from 1–several 10 μm in diameter, e.g., Ishizuka et al., 2017; Kinase et al., 2018); thus, the deposition parameters should be much larger. In this study, $E_c$ and $v_d$ are significantly improved from those used in K16, as is extensively described later in Sect. 2.3.

*2.2.2 Resuspension schemes*

K16 considered three emission sources during the analysis period of 2013: resuspension from bare soil, resuspension from forest ecosystems, and additional emissions from the reactor buildings of the F1NPP. K16 simulated the contributions from these additional emissions as being two to three orders of magnitude smaller than the observed surface activity concentrations, and these contributions were thus neglected in this study. The emission flux of $^{137}$Cs carried by dust aerosols from a bare soil
surface, $F_{dust}$ (Bq m$^{-2}$ s$^{-1}$), is formulated by Ishizuka et al. (2017) as follows:

$$F_{\text{dust}} = p_{20\mu m}F_{\text{M}}(1 - f_{\text{forest}})B_{5\text{mm}}(t)C_{\text{const}}, \tag{3}$$

where $p_{20\mu m}$ is the surface area fraction of dust particles smaller than 20 μm in diameter against soil containing a maximum particle size of 2 mm and varies depending on the soil texture ($1.3\times10^{-8}$ for sand, 0.19 for loamy sand, 0.45 for sandy loam, and 0.80 for silt loam), $F_M$ is the total dust mass flux (kg m$^{-1}$ s$^{-1}$) as a function of the friction velocity, $f_{forest}$ is the forest areal fraction, and $B_{5mm}(t)$ is the specific radioactivity of the surface soil (from the surface to a depth of 5 mm; Bq kg$^{-1}$) as a function
of time considering radioactive decay. Changes in the vertical profiles of $^{137}$Cs due to land surface processes or decontamination are not considered in the study. The $f_{forest}$ value is obtained from the Weather Research and Forecasting model version 3 database (WRFV3; Skamarock et al., 2018). Eq. 3 was developed based on measurements taken in a schoolyard, so it may not be applicable for every soil surface type. For simplicity, we introduce the constant correction factor $C_{const}$ to adjust





the simulated $^{137}$Cs in dust aerosols to the observed value. $C_{const}$ was set to five in K16. This adjustment factor differs in this

study because a larger adjustment factor is required to sustain the observed surface concentration levels for faster deposition rates, as shown later in Sect. 2.3.

The resuspension flux of $^{137}$Cs from forest ecosystems (regarded as forest aerosols), $F_{forest}$ (Bq m$^{-2}$ s$^{-1}$), is formulated by K16 as follows:

$$F_{forest} = f_{forest}f_{green}r_{const}B_{obs}R_{decay}(t), \tag{4}$$

where $f_{green}$ is the monthly mean green area fraction, $r_{const}$ is the constant resuspension coefficient (s$^{-1}$), $B_{obs}$ is the observed

initial deposition amount (Bq m$^{-2}$, Fig. 1b), and $R_{decay}(t)$ is the radioactive decay. $f_{green}$ is obtained from the WRFV3 database and was originally derived from Advanced Very High Resolution Radiometer (AVHRR) normalized difference vegetation index (NDVI) data. $r_{const}$ is the adjustment parameter and was set as $10^{-7}$ h$^{-1}$ in K16 for adjustment to the observed surface concentrations in the forests in summer. Similar to the dust aerosol case, a larger adjustment factor is required due to the faster deposition rates to sustain the simulated surface concentrations close to the observed values, as is shown later in Sect. 2.3.

Similar to the dust aerosol case, no $^{137}$Cs migration within the local forest ecosystems due to land surface processes is considered in the formulation.

For both the dust and forest aerosol cases, only emissions from the grids in which the mean initial deposition amounts exceed 10 kBq m$^{-2}$, the detection limit of the airborne measurements, are considered (NRA, 2012). However, excluding regions in which the deposition amount is 9.9 kBq m$^{-2}$, for example, may not be appropriate. Thus, a sensitivity test is performed to

additionally consider areas with deposition amounts of 1–10 kBq m$^{-2}$ as emission sources, as is presented in Sect. 3.4.

Other sources, such as the unexpected releases associated with the debris removal operations at the F1NPP site that occurred in August 2013 (NRA, 2014; Steinhauser et al., 2015; K16), forest fires, and resuspension due to decontamination work, are not considered in the study. The debris removal operations caused a sporadic peak in the surface concentrations (60.4 mBq m$^{-3}$ from 13:00 LT on August 14 to 13:00 LT on August 15 at Namie (Tsushima), Figs. 4a and 4b), but these

elevated values may not affect the background (or steady state) concentrations for the full year, which are the target of this study. Forest fires may not be a major source of $^{137}$Cs resuspension in Fuskuhima because the temporal variations in levoglucosan concentrations were not found to be associated with those of $^{137}$Cs (Kinase et al., 2018). Resuspension due to decontamination work should be considered, but it was hard to estimate because the emission factor and the precise location and time of decontamination are unknown. It should be noted here that, as described in Sect. 2.1, decontamination was not

performed around the Namie (Tsushima) or Tsukuba sites, and decontamination might have been performed around the Fukushima site in 2013.


**2.3 Constrained deposition parameters and emission flux adjustments based on field observation data**

*2.3.1 Constraint of modeled deposition parameters*

Since K16 was published, several emission sources of resuspended $^{137}$Cs have been indicated, such as soil dust (Ishizuka et al., 2017; Kinase et al., 2018) and bioaerosols (Kinase et al., 2018; Nakagawa et al., 2018; Igarashi et al., 2019b; Kita et al., 2020; Minami et al., 2020; Igarashi, 2021), but the relative contributions of these sources, the spatiotemporal variations in the associated emission fluxes, and their size distributions are still not well understood. Kita et al. (2020) indicated the associations of rain with fungal spore emissions, and Minami et al. (2020) estimated the emission flux of $^{137}$Cs associated with bioaerosols; however, the emission flux has not yet been formulated as a function of meteorological or land surface

variables. Therefore, the same formulations as those applied in K16 (Eqs. 3 and 4) are used in this study. It is also noted here that the same deposition rates are applied for both dust and forest aerosols, even though the size distributions of these two aerosol types should be different.

     K16 had two major drawbacks: (1) K16 constrained the deposition parameters (i.e., $E_c$ and $v_d$) by using the primary $^{137}$Cs emission and initial $^{137}$Cs deposition amounts measured in March 2011, and (2) K16 did not compare their simulation

results against the deposition amounts. Recently, Watanabe et al. (2021) evaluated the performance of the K16 model using concentrations and deposition amounts measured at Fukushima sites and found that the seasonal variations in simulated concentrations were opposite to those observed and that the simulated deposition amounts were underestimated by one to two orders of magnitude. The reason for this underestimation of the deposition amounts is obvious; the typical deposition rates of major carrier aerosols (submicron aerosols) are much smaller than the resuspension rates (supermicron aerosols). For example,

the dry deposition velocities of aerosols with diameters of approximately 10 μm are two to three orders of magnitude larger than those of aerosols with diameters of approximately 0.1–1 μm (e.g., Petroff and Zhang, 2010). The difference between these two size ranges for below-cloud scavenging due to rain is also two to three orders of magnitude (e.g., Wang et al., 2010). To constrain the deposition parameters suitable for $^{137}$Cs resuspension, we performed a climatological deposition velocity analysis similar to that conducted by Watanabe et al. (2021).

Suppose there is a simple nonlinear relationship between the periodic mean deposition flux ($D$) (Bq m$^{-2}$ s$^{-1}$, for example) and periodic mean surface concentration ($C$) (Bq m$^{-3}$):

$$D = aC^b,  \hspace{2cm} (5)$$

where $a$ represents the removal rate and $b$ represents nonlinear features such as spatial and temporal variabilities. If $b = 1$, the unit of $a$ is m/s, which is on the dimension of the deposition velocity. If long-term averaging is conducted, Eq. 5 may hold. Eq. 5 is reformulated as follows:



$$log(D) = b \, log(C) + log(a). \tag{6}$$

A log-log scatter plot between the monthly mean surface concentrations and monthly cumulative depositions is shown in Fig. 2. The purple, orange, and green symbols indicate the observations, simulations by K16 ($E_c$ = 0.04 and $v_d$ = 0.1 cm s$^{-1}$) and simulations conducted in this study ($E_c$ = 0.4 and $v_d$ = 10 cm s$^{-1}$). This analysis is novel because emission flux adjustments ($C_{const}$ in Eq. 3 and $r_{const}$ in Eq. 4) do not change the slope of the regressions, so the deposition parameters can be adjusted independently from the emission flux adjustment. The intercept of the y-axis indicates the deposition velocity. Among the

several sensitivity tests with the combinations of $E_c$ = 0.04 and 0.4 and $v_d$ = 0.1, 1, and 10 cm s$^{-1}$, the y-axis intercept of the simulation with $E_c$ = 0.4 and $v_d$ = 10 cm s$^{-1}$ matched best with that of the observations.

      The slope of the observed regression line, $b$, is 0.92, so the relationship between the concentrations and depositions of resuspended $^{137}$Cs in East Japan is almost linear, but the relationship itself is not very solid (coefficient of determination ($R^2$) = 0.018). However, by excluding two exceptional data points, the observations obtained in January at Fukushima

(maximum deposition) and August at Namie (Tsushima) (maximum concentration), the $R^2$ increases to 0.65, and $b$ becomes 0.98 (not shown in the figure). In January at Fukushima, the measured deposition is extremely high compared to the surface concentration. Watanabe et al. (2021) hypothesized the existence of superlarge particles (~100 μm in diameter) whose gravitational deposition velocities are too fast (as fast as drizzle) to enter the high-volume air samplers used for the concentration measurements but are efficiently collected deposition samplers, as the traveling distance is approximately 1 km

(e.g., Kajino et al., 2012; Kajino et al., 2021). January is the month with the highest snow cover in Fukushima city and the highest snow removal operations (using snowblowers and deicing agents), and heavy traffic on the major roads within 1 km of the Fukushima site produce substantial amounts of superlarge particles from wet and muddy road surfaces. The August data at Namie (Tsushima) are also exceptional because the surface concentrations are biased due to the sporadic peak associated with the debris removal operation in the F1NPP (K16). Because the aerosols associated with the debris removal operation

traveled a sufficiently long distance (i.e., 30 km), the deposition velocity was not significantly large and did not affect the monthly mean deposition, although it did affect the monthly mean concentration. Therefore, the data obtained in August at Namie (Tsushima) are exceptional when compared to the trends shown by other datasets. The slopes of the simulated regression lines are $b$ =1.17 for K16 (orange line) and $b$ =1.16 for this study (green line). $R^2$ values of 0.71 and 0.97 were obtained by K16 and this study, respectively. The difference in the magnitude of $R^2$ can be explained by the differences in the deposition

rates. Because the deposition rates obtained in this study are much faster than those applied by K16, the deposition amounts are more strongly associated with the concentrations in this study. In other words, this climatological deposition velocity analysis was successful (by excluding the two exceptional datapoints) because the sizes of the major carrier aerosols of resuspended $^{137}$Cs in reality are sufficiently large (the observed deposition amounts are sufficiently associated with the observed concentrations). The regression slopes of the simulations ($b$ ~ 1.2) are somewhat different from those observed ($b$ ~

0.9 or 1.0). Nevertheless, the regression line of this study crosses that of the observations at the middle points of the





concentration and deposition ranges (approximately 0.1 mBq m$^{-3}$ and 50 Bq m$^{-2}$, respectively). This indicates that the constrained deposition rates may be consistent with the average features of the environmental behaviors of resuspended $^{137}$Cs in East Japan.

This analysis is conventional but has been found to be quite successful in constraining the deposition rates of resuspended $^{137}$Cs in East Japan.

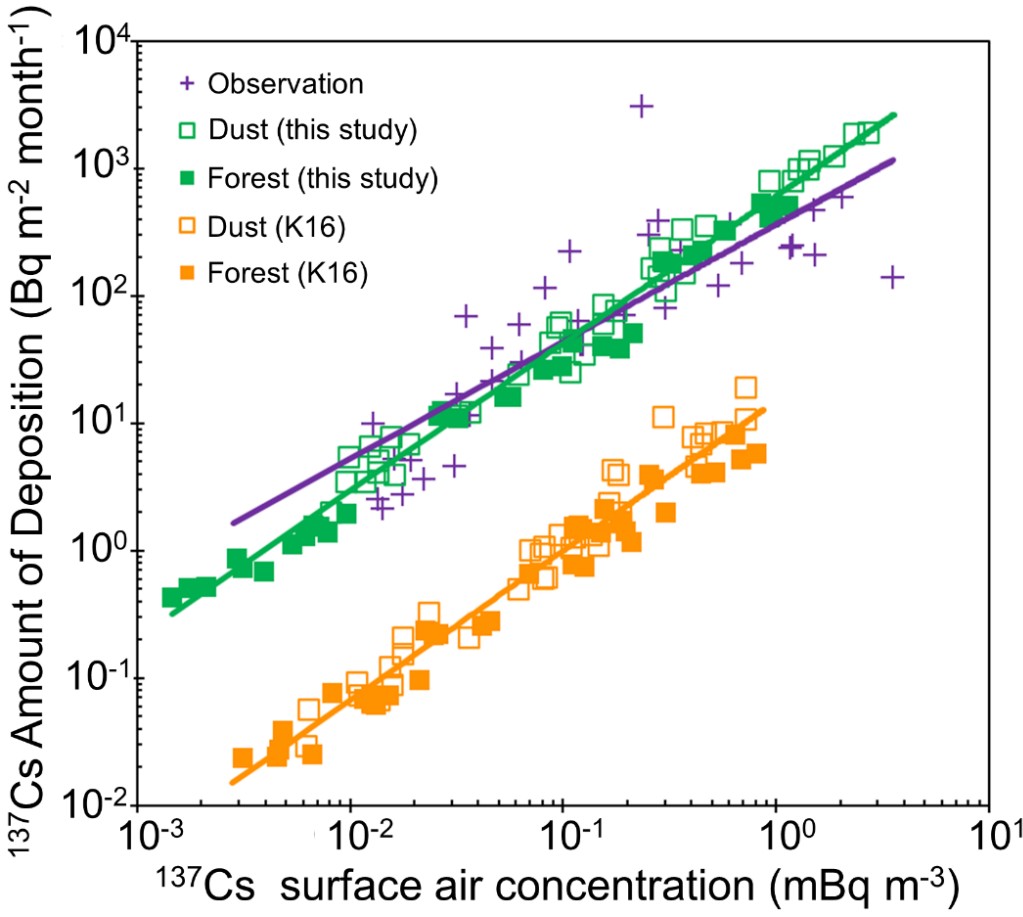

**Figure 2:** Scatter diagram of the depositions of $^{137}$Cs over the monthly mean (purple crosses) observed surface concentrations at Namie (Tsushima), Fukushima, and Tsukuba in 2013 and those simulated by Kajino et al., 2016 (K16) ($E_c$ and $v_d$ are 0.04 
and 0.1 cm s$^{-1}$, respectively) considering different emission sources: the open orange squares represent mineral dust particles from bare soil (dust aerosols), and the closed orange squares denote bioaerosols emitted from forest ecosystems (forest aerosols). The green open and closed squares are the same as the orange squares but are simulated by this study ($E_c$ and $v_d$ are 0.4 and 10 cm s$^{-1}$, respectively). The purple, orange, and green lines indicate the regression lines of the purple crosses, orange squares (open plus closed), and green squares (open plus closed), respectively.



*2.3.2 Adjustment of emission fluxes*

In K16, the emission fluxes of $^{137}$Cs associated with dust and forest aerosols were adjusted to match the surface concentrations at Namie (Tsushima). In K16, first, $C_{const}$ in Eq. 3 was set to five so that the simulated dust $^{137}$Cs concentrations matched the observations in winter, when the temporal variation in the observed $^{137}$Cs concentration at Namie (Tsuhima) correlated well with that of the wind speed (Kinase et al., 2018) and when the vegetation activity was supposed to be low. The

adjusted dust $^{137}$Cs concentrations could not reproduce the enhanced concentrations measured at Namie (Tsushima) in summer (Fig. 4a). Thus, $r_{const}$ in Eq. 4 was set to $10^{-7}$ h$^{-1}$ so that the simulated forest $^{137}$Cs concentrations matched the observations in summer. The temporal variation of $^{137}$Cs was not correlated with that of the wind speed in summer (Kinase et al., 2018).

Because the deposition rates are significantly increased in this study, we require much larger emission fluxes to sustain the simulated surface concentrations at the observed levels. The same adjustment procedure as that used in K16 could be

applied to the simulations in this study; however, for example, adjusting the values at Namie causes the values to be underestimated at Tsukuba, so it is hard to find a combination of $C_{const}$ and $r_{const}$ that is best for all aspects (i.e., concentrations and depositions of the three sites). Thus, for simplicity, we multiplied both fluxes used in K16 by 20 so that $C_{const}$ was 100 and $r_{const}$ was $2\times10^{-6}$ h$^{-1}$. The discrepancies between the simulated and observed concentrations and depositions at the three sites are summarized later in Sect. 3.1.

**3 Results and discussion**

**3.1 Seasonality and quantity of surface air concentrations and depositions**

Figure 3 shows the observed and simulated (dust and forest) activity deposition amounts of $^{137}$Cs at Namie (Tsushima), Fukushima, and Tsukuba for the submicron (K16) and supermicron (this study) cases. Statistical measures such as the correlation coefficient ($R$), simulation-to-observation median ratio ($Sim/Obs$), numerical fraction of data within a factor of two

($FA2$), and numerical fraction of data within a factor of five ($FA5$) are embedded in the panels. Note that the simulated temporal variation lines show the dust and forest amounts separately, but the statistical measures are derived using the summation of the two aerosol sources. As discussed in the previous sections and presented in Fig. 2, the underestimation of simulations assuming submicron particles is remarkable, with simulated values approximately two orders of magnitude lower than the observations at all sites. On the other hand, the simulations assuming supermicron particles are significantly improved. Positive

correlations are found at Namie (Tsushima) and Tsukuba ($R \sim 0.6$–$0.7$), and the same order of median ratios are found at Fukushima and Tsukuba ($Sim/Obs = 1.2$–$1.3$, $FA5 = 0.9$–$1.0$). In terms of the seasonal variations, the monthly trend (high in winter and spring and low in summer) at Tsukuba is explained well by the simulated dust aerosols. Due to the land use types around the site (over the plain), the $^{137}$Cs of dust aerosols is larger than that of forest aerosols throughout the year. In summer, the contributions of forest aerosols are larger than those of dust aerosols at Namie (Tsushima) and Fukushima, which are




surrounded by mountainous forest and close to the forest area, respectively. Despite the overestimation at Namie (Tsushima) (*Sim/Obs* = 4.8), the monthly trend is reproduced well by the model: both the observations and simulations show double peaks in winter and summer. Most likely, the same emission factors of mineral dust should not be applied to the whole area. Nevertheless, we regard this application as acceptable in the current study, as this study aims to grasp a rough outline of the atmospheric behaviors of resuspended $^{137}$Cs. The monthly variations output by the simulations do not match those of the observations at Fukushima due to the exceptionally high deposition amounts observed in January. This is possibly due to the existence of superlarge particles, as described in Sect. 2.3.1. The snow coverage in Fukushima city is highest in January, but there are certain areas of snow coverage in December and February as well. However, the *R* value of deposition at Fukushima is not greatly improved if the winter datasets are excluded.

The initial $^{137}$Cs depositions at the three sites are 2300, 190, and 21 kBq m$^{-2}$, and the differences are approximately on one order of magnitude. The orders of the monthly depositions in 2013 at the three sites are 0.1% of the initial deposition, at $10^2$-$10^3$, approximately $10^2$, and approximately $10^1$ Bq m$^{-2}$, which are similar to the order differences obtained for the initial depositions. A value of 0.1% per month is 1% per year. K16 reported an annual resuspension ratio of 0.048% y$^{-1}$, but this simple order estimation readily shows that this value is excessively underestimated. As shown later in Fig. 8, the improved annual resuspension ratio is 0.96%, which is consistent with the deposition measurements at the three sites. From this estimation, one can assume that the observed deposition amounts at Fukushima in January (3100 Bq m$^{-2}$) are exceptionally high.





**Figure 3:** Monthly deposition amounts of (black) observed $^{137}$Cs and simulated $^{137}$Cs associated with (red) dust aerosols and (lime) forest aerosols at Namie (Tsushima), Fukushima, and Tsukuba (Bq m$^{-2}$). The simulation results assuming submicron particles (K16; $E_c$ and $v_d$ are 0.04 and 0.1 cm/s, respectively) and those assuming supermicron particles (this study; $E_c$ and $v_d$ are 0.4 and 10 cm/s, respectively) are shown on the left and right, respectively. The statistical measures, such as the correlation coefficient ($R$), simulation-to-observation median ratio (*Sim/Obs*), numerical fraction of data within a factor of two (*FA2*), and numerical fraction of data within a factor of five (*FA5*), between the observations and the simulations (dust plus forest) for each simulation result are embedded in each panel.







**Figure 4:** Same as Fig. 3 but for the surface activity concentrations of $^{137}$Cs (mBq m$^{-3}$). The sampling intervals are used for the observations, but daily mean values are depicted for the simulations.


Figure 4 shows the observed and simulated (dust and forest) surface air activity concentrations of $^{137}$Cs at the three sites for the submicron (K16) and supermicron (this study) cases. The statistical measures $R$, *Sim/Obs*, *FA2*, and *FA5* between the observations and simulations (dust plus forest) are also embedded in the panels. The lines are depicted using different temporal resolutions (sampling intervals for the observations and daily for the simulations), but the temporal resolutions are unified to the sampling intervals to obtain the statistical measures. The measurements of the three sites are not directly comparable because their temporal resolutions are different (1 d for Namie (Tsushima), 2–3 d for Fukushima and 1 w for Tsukuba), but those of the two aerosol cases (submicron and supermicron) are comparable. Although $R$ is low for the submicron



case at Namie (Tsushima), good consistency *Sim/Obs* (0.99) and *FA5* (0.94) values are obtained because the emission factors $C_{const}$ and $r_{const}$ are adjusted to this case. However, the unrealistic assumption of aerosol sizes results in the opposite simulated

seasonal trend at Fukushima: the simulations are high in summer due to forest aerosols. The Fukushima site is located downwind of the contaminated forest in the Abukuma Highlands in summer (Fig. S1), so the transport of $^{137}$Cs from the forest area is dominant. However, as the particle sizes are larger and the traveling distances are shorter, the summer enhancement due to forest aerosols is less dominant (Fig. 4d). The observed surface concentrations are high at Fukushima in winter, and the observed short-term peaks correspond to the simulated dust aerosols, indicating that the emission of resuspended $^{137}$Cs at

Fukushima in winter is driven by wind. The *Sim/Obs*, *FA2*, and *FA5* values of submicrons and supermicrons at Fukushima are similar, but *R* is substantially improved. At Tsukuba, like the Fukushima site, the contribution of forest aerosols is less in the supermicron case than in the submicron case due to less transport from the forest area. The contribution of dust particles is dominant in winter, but the simulated dust aerosols are underestimated compared to the observations in winter. Nevertheless, *Sim/Obs* is not very low (0.80), and the *R* value obtained for supermicrons is improved from the submicron case (from 0.18 to

400 0.45).

The orders of the surface concentrations at the three sites are $10^0$, $10^{-1}$, and $10^{-2}$-$10^{-1}$ mBq m$^{-3}$. These order differences are similar to those of the initial depositions. One can assume that the resuspension and redeposition of $^{137}$Cs occurs within a limited areal scale (e.g., several tens of km) and that long-range transport (i.e., hundreds to a thousand km) from the emission source is not very dominant.

In the following subsections (Sects. 3.2 and 3.3), the source-receptor relationship and annual resuspension ratios are discussed, but it should be noted that the numbers presented in these sections are associated with the discrepancies in the simulations described in the current section. Nevertheless, we can safely conclude here that the supermicron simulations are more (or maybe much more) consistent with the observations than the submicron simulations are.




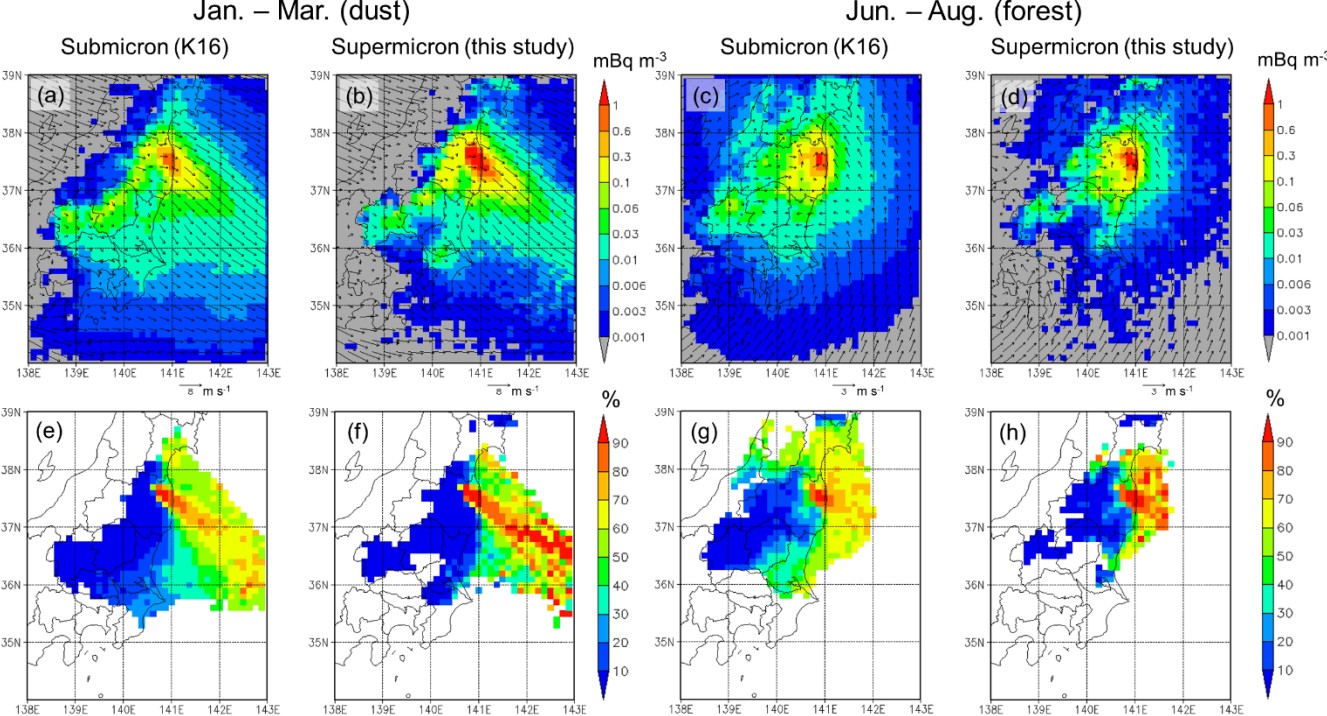

**Figure 5:** (top panels) Seasonal mean surface concentrations of (a,c) submicron (K16; $E_c$ and $v_d$ are 0.04 and 0.1 cm s$^{-1}$, respectively) and (b,d) supermicron (this study; $E_c$ and $v_d$ are 0.4 and 10 cm s$^{-1}$, respectively) $^{137}$Cs associated with (a,b) dust aerosols in winter to early spring (January, February, and March) and (c,d) forest aerosols in summer (June, July, and August) (mBq m$^{-3}$). The seasonal mean surface wind vectors are also depicted in the panels. (bottom panels) Same as top panels but for the fractional contributions from the resuspension source area (defined as the initial depositions of $^{137}$Cs exceeding 300 kBq m$^{-2}$, see Fig. 1b) to the surface concentrations (%).

## 3.2 Source-receptor relationship and its seasonality

Figure 5 shows the simulated seasonal mean concentrations and the horizontal distributions of the source-receptor relationship. The resuspension source area is defined as grids in which the grid-mean initial deposition exceeds 300 kBq m$^{-2}$ (Fig. 1b). Thus, Namie (Tsushima) (2300 kBq m$^{-2}$) is located within the source area, but Fukushima (190 kBq m$^{-2}$) and Tsukuba (21 kBq m$^{-2}$) are outside the source area (or are regarded as being in downwind area). The source-receptor relationship maps (or source contribution maps) (Figs. 5e–5h) are derived using the seasonal mean concentrations from 300-kBq m$^{-2}$ areas divided by those from whole areas (i.e., > 10 kBq m$^{-2}$). Because of the shorter atmospheric lifetime of supermicron $^{137}$Cs-bearing particles, the concentration maps of the supermicron cases are patchy due to insufficient amounts of Lagrangian particles (Figs. 5b and 5d) compared to the submicron cases (Figs. 5a and 5c), especially in areas where the seasonal mean surface concentrations are below 0.01 mBq m$^{-3}$. There are substantial numerical errors in these areas, so the source contribution shades (Figs. 5e–5h)





depict only areas in which the seasonal mean concentrations exceed 0.01 mBq m$^{-3}$ (Figs. 5a–5d). We select two three-monthly
means, covering January, February, and March for winter to early spring (or simply winter hereinafter) when the simulated
dust aerosols are dominant and June, July, and August for summer when the simulated forest aerosols are dominant.

In winter, northwesterly monsoon winds prevail over Fukushima Prefecture (Figs. 5a and 5b). In particular, fall and
gap winds from the Ou Mountains caused strong winds in the Nakadori Valley, which in turn cause high dust aerosol surface
concentrations in these areas in winter. Even though the surface concentrations of supermicron particles over Fukushima
Prefecture are larger than those of submicron particles (Fig. 5b), the supermicron concentrations over the downwind regions
are smaller (e.g., concentrations > 0.01 mBq m$^{-3}$ over Saitama (#6 in Fig. 1) for the submicron case but of < 0.01 mBg m$^{-3}$ for
the supermicron case) due to the shorter lifetime of supermicron particles. This feature is also significant for the source
contribution maps (Figs. 5e and 5f). Due to the northwesterlies, most of the resuspended $^{137}$Cs is transported toward the
southeast over the ocean, the values are 40–90%, and the source contributions of the downwind regions over the land are lower
than 10%, except the coastal regions in Ibaraki (#3 in Fig. 1) and Chiba (#7 in Fig. 1) Prefectures for the submicron case (20–
30%) due to the longer lifetimes of these particles in air (Fig. 5e).

In summer, southerly winds prevail over East Japan due to the marginal flows of the Pacific High. The wind speeds
are generally lower in summer than in winter (please see that the lengths of the arrows are different in Figs. 5a–5b and 5c–5d).
The seasonal mean wind patterns are complex over land (Figs. 5c–5d), but the seasonal mean source contribution maps reflect
the seasonal mean transport patterns from the source areas (Figs. 5g–5h). Even though the seasonal mean wind fields over the
ocean close to land are directed toward the land, substantial proportions of $^{137}$Cs in forest aerosols are transported toward the
ocean in summer (the source contributions are > 60% for submicron and >70% for supermicron cases). Then, the $^{137}$Cs
transported toward the ocean are transported toward the land again to Ibaraki and Miyagi (#1 in Fig. 1) prefectures. The source
contributions over Ibaraki and Miyagi are substantial for the submicron case (30–70%). For the supermicron case, the source
contributions over Ibaraki and Miyagi exceeded 30% at a limited number of grids, but the mean concentrations were much
lower (Fig. 5d) than those in the submicron case (Fig. 5c) over the prefectures. Due to the lower wind speeds in summer and
the short lifetime of the supermicron particles, the horizontal spread of the mean concentrations (e.g., areas > 0.01 mBq m$^{-3}$)
of supermicron forest aerosols in summer (Fig. 5d) is obviously smaller than that of any other case (Fig. 5a–5c).





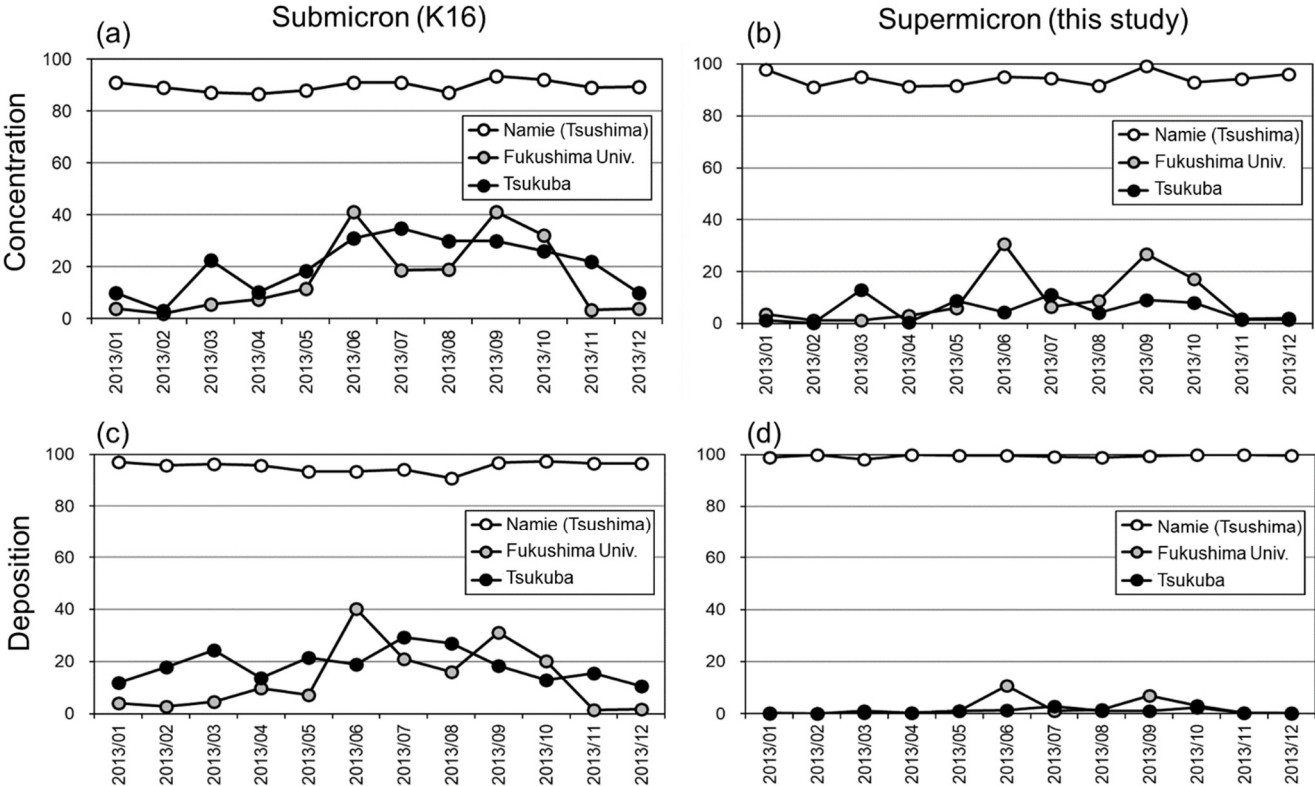

**Figure 6:** Monthly mean fractional contributions from the resuspension source area (defined as initial depositions of $^{137}$Cs exceeding 300 kBq m$^{-2}$) to the (a,b) surface concentrations and (c,d) deposition amounts assuming (a,c) submicron (K16; $E_c$ and $v_d$ are 0.04 and 0.1 cm s$^{-1}$, respectively) and (b,d) supermicron (this study; $E_c$ and $v_d$ are 0.4 and 10 cm s$^{-1}$, respectively) sizes of $^{137}$Cs-bearing particles at Namie (Tsushima), Fukushima, and Tsukuba.

Figure 6 shows the simulated monthly mean source contributions for the concentrations and depositions at the three sites and compares these contributions between the submicron and supermicron cases. At Namie (Tsushima), more than 90% of the surface concentrations originate from the source area. The annual mean values are 90% for the submicron case (Fig. 6a) and 94% for the supermicron case (Fig. 6b). It is natural that the source contributions are larger in the source area (Namie (Tsushima)) for the supermicron case, as the lifetime of these particles is shorter than that of submicron particles. As previously discussed, the submicron source contributions at Fukushima are larger in summer and autumn (approximately 40% in June, September, and October and 20% in July and August) (Fig. 6a). The source contributions of supermicrons in summer and autumn are approximately 50% smaller than those of submicrons in July and August (10%) and are slightly smaller in June, September, and October (20–30%) (Fig. 6b). The annual mean concentration values at Fukushima are 16% for submicrons (Fig. 6a) and 9% for supermicrons (Fig. 6b). As observed in Figs. 4e–4f and Figs. 5e–5h, the source contributions of submicrons and supermicrons at Tsukuba are remarkably different. The source contributions of submicrons are larger in



summer and autumn at approximately 30%, with an annual mean value of 21% (Fig. 6a). On the other hand, those of supermicrons are smaller than 20% for all months, and the annual mean value is 5% (Fig. 6b).

In terms of deposition, the source contributions of submicrons (Fig. 6c) are similar to those for the concentrations (Fig. 6a), but the source contributions of supermicrons (Fig. 6d) are remarkably different. As presented later in Fig. 7,
approximately 40% of supermicron emissions were deposited in the same grid. Thus, the local deposition contributions become much larger than the concentration contributions. Consequently, as shown in Fig. 6c, the annual mean source contributions are 95%, 13%, and 19% for Namie (Tsushima), Fukushima, and Tsukuba, respectively, which are equivalent to those of the concentrations (90%, 16%, and 21% in Fig. 6a), whereas in Fig. 6d, the annual mean source contributions of supermicron depositions are shown to be 99.5%, 2.2%, and 1.0% for Namie (Tsushima), Fukushima, and Tsukuba, respectively.

**3.3 Annual total resuspension amounts**

Figure 7 presents the simulated annual resuspension and redeposition amounts for the submicron and supermicron cases. The areal summation values are embedded in the panels. Figure 8 shows the annual resuspension ratio, which is the annual total resuspension amount divided by the initial deposition (Fig. 1b), and the annual redistribution amount, which is the deviation between the annual total redeposition amount and the resuspension amount. Negative redistribution values indicate a decrease
in deposition due to resuspension, and positive values indicate an increase in deposition due to resuspension. The annual total amounts embedded in Figs. 8b and 8d (-1.06 TBq and -15.1 TBq) indicate the amount of $^{137}$Cs transported outside the model domain.

As previously discussed, due to the faster deposition rates and thus larger emission fluxes necessary to sustain the surface concentrations at the observed levels, the annual resuspension and redeposition amounts are both larger for the
supermicron case than for the submicron case (Figs. 7c–7d). In K16, the total areal resuspension amount was 1.28 TBq (Fig. 7a), equivalent to only 0.048% of the initial deposition (2.68 PBq), and the redeposition amount was approximately 20% of the resuspension amount. On the other hand, based on the new estimations, the annual resuspension amount (25.7 TBq) is approximately 20 times the previous estimate, and the redeposition amount is even larger (10.6 TBq), at 50 times the previous estimate (0.22 TBq).

The areal mean annual resuspension ratio obtained by K16 (Fig. 8a) was 0.048%, with high values above 0.1% in Nakadori Valley and the mountainous areas of Tochigi (#4 in Fig. 1) and Gunma (#5 in Fig. 1) Prefectures. The new estimate of the annual mean areal resuspension ratio is 0.96%, with high-value areas showing values of 1–3% (Fig. 8c). Iwagami et al. (2017) evaluated that the annual discharge rate from the local environment through rivers was 0.02–0.3% y$^{-1}$. The new estimate of the resuspension rate through air (0.96% y$^{-1}$) is much larger than the discharge rate through rivers but is still not very large
(i.e., only 1% per year of surface contamination). We can conclude here that the ground surface $^{137}$Cs stays or circulates within the local terrestrial ecosystems and is hardly discharged through the air or rivers.





However, when these values are compared with the actual decreasing trends in the ambient gamma dose rate in Fukushima Prefecture, we can reach a different conclusion. The first-order decrease rate of the dose rate in Fukushima Prefecture ranged from 1.0 to $7.9 \times 10^{-4}$ $d^{-1}$ in 2013 after the radioactive decay of $^{137}$Cs plus $^{134}$Cs ($3.0-4.2 \times 10^{-4}$ $d^{-1}$ in 2013)

(K16) was subtracted. The previous and current resuspension rate estimates, 0.048% $y^{-1}$ and 0.96% $y^{-1}$, are equivalent to $1.3 \times 10^{-6}$ $d^{-1}$ and $2.6 \times 10^{-5}$ $d^{-1}$, respectively. K16 concluded that the impact of resuspension to the atmosphere was negligibly (two to three orders of magnitude) small in the gross decreasing rate of ground surface contamination, such as by land surface processes or decontamination. However, the difference between the current estimate ($2.6 \times 10^{-5}$ $d^{-1}$) and the decreasing trend without radioactive decay ($1.0-7.9 \times 10^{-4}$ $d^{-1}$) is only one to two orders of magnitude. The annual resuspension ratio in 2013

was only 0.96% of the initial deposition amount, but the amount of $^{137}$Cs discharged due to resuspension through the air could contribute approximately 1–10% of the gross decreasing rate, which may not be negligibly small.

For the submicron case (Fig. 8b), the positive redistribution area (area enhanced deposition due to resuspension) is limited, and the amounts are up to 10 Bq $m^{-2}$ per year. On the other hand, for the supermicron case (Fig. 8a), even though the transport distance is shorter than that for submicrons, the positive redistribution area for 1–10 Bq $m^{-2}$ is much larger, and the

maximum values are up to 100 Bq $m^{-2}$ for the downwind regions close to the emission sources, especially over the ocean close to the land of Fukushima Prefecture. Nevertheless, these values are much smaller than those obtained for the initial deposition amounts (the lowest limit value is 10 kBq $m^{-2}$, which is two to three orders of magnitude larger than the annual enhanced deposition amounts of 10–100 Bq $m^{-2}$).





**Figure 7:** Horizontal distributions of (a,c) the annual total amounts of resuspended $^{137}$Cs (Bq m$^{-2}$) and (b,d) redeposited amounts of resuspended $^{137}$Cs (Bq m$^{-2}$) obtained from the simulations assuming (a,b) submicron (K16; $E_c$ and $v_d$ are 0.04 and 0.1 cm s$^{-1}$, respectively) and (c,d) supermicron (this study; $E_c$ and $v_d$ are 0.4 and 10 cm s$^{-1}$, respectively) sizes of $^{137}$Cs-bearing particles. The areal total amounts are embedded at the bottom right of each panel.



**Figure 8:** Horizontal distributions of (a,c) the annual total resuspension ratio (ratio of resuspension amounts to initial deposition amounts) of [137]Cs (%) and (b,d) the annual redistribution (redeposited minus resuspended amounts) of [137]Cs (Bq m[-2]) obtained from simulations assuming (a,b) submicron (K16; $E_c$ and $v_d$ are 0.04 and 0.1 cm s[-1], respectively) and (c,d) supermicron (this study; $E_c$ and $v_d$ are 0.4 and 10 cm s[-1], respectively) sizes of [137]Cs-bearing particles. The areal total amounts ratios (a,c) and the areal total amounts (b,d) are embedded at the bottom right of the panels.





### 3.4 Sensitivities

Several sensitivity tests are performed, as shown in the current section. Since the cumulus convection parameterization scheme is installed, a comparison is made between the simulations performed with (in the current study) and without (in K16) this scheme. The current study assumes that resuspension occurred from the grids in which the grid-mean initial deposition amount exceeded 10 kBq m$^{-2}$, the reliable limit of the aircraft measurement. On the other hand, it is inappropriate to exclude grids in which, for example, the deposition amount was 9.9 kBq m$^{-2}$, so additional sensitivity tests include resuspension from grids

with 1–10 kBq m$^{-2}$. As was discussed in part in the previous sections, the snow cover effect is also tested. In summary, for each aerosol size case, we conduct four sensitivity tests: (1) No cumulus parameterization, denoted as [No. cuml.], (2) with cumulus parameterization [Cuml.], (3) [Cuml.] plus the inclusion of resuspension from 1–10-kBq m$^{-2}$ grids [1–10kBg m$^{-2}$], and (4) [Cuml.] plus [1–10 kBq m$^{-2}$] plus the snow cover effect [Snow cover]. Thus, the submicron case of [No cuml.] was used in the study of K16, and the supermicron case of [Cuml.] is used as the simulation in this study. Note that the submicron

simulations shown in the current study are [Cuml.]. While this parameterization is complicated, the differences between [Cuml.] and [No cuml.] are exceedingly small.

Figure 9 presents monthly mean snow cover data interpolated to the model grids. The original data are the MODerate resolution Imaging Spectroradiometer (MODIS) snow cover collection 6 level-3 data (MOD10CM, global, monthly, 0.05° resolution) (Riggs et al., 2016). In the presence of snow cover, the simulated dust emission is suppressed by the snow cover

fraction (Eq. 3 is multiplied by one minus snow cover). No impact of snow cover on forest emissions is assumed in the simulation. December, January, and February are the months with the widest snow coverage in East Japan. In November, only small snow cover fractions are observed in the high-mountain areas (i.e., > 1000 m in Fig. 1a). In March, the snow cover in the low-elevation areas (i.e., < 1000 m in Fig. 1a and over all prefectures numbered in Fig. 1b except the western part (Ou Mountains) of Fukushima Prefecture) is mostly melted. The snow cover in the Nakadori valley, including at the Fukushima

site, is highest in January. Some areas over the Kanto Plain are covered with snow in January. Extensive snow cover is also observed in the Abukuma Highlands, including in Namie (Tsushima), in February.

Figure 10 compares the statistical metrics $R$, $Sim/Obs$, $FA2$, and $FA5$ of the concentrations (daily to weekly depending on the site) and monthly depositions for the eight sensitivity tests. As previously discussed, the statistical scores of the supermicron simulations are significantly greater than those of the submicron simulations, especially the $R$ values of the

concentrations at Fukushima and Tsukuba, the $R$ value of deposition at Tsukuba, the $Sim/Obs$ values at Fukushima and Tsukuba, and the $FA2$ and $FA5$ values of deposition at all sites. Including the cumulus parameterization was successful in the sense that it did not cause any significant deterioration in the statistical scores. The supermicron simulations indicate slight improvements due to cumulus convection, such as increased $R$ values of the concentrations at Fukushima and Tsukuba, but the $FA2$ values of the concentrations at Fukushima and Tsukuba are slightly decreased. The impact of including a 1–10-kBq m$^{-2}$ area would



be larger at Tsukuba, which is surrounded by less contaminated regions. The supermicron simulations indicate a slight improvement in the *R* value of the concentrations at Tsukuba, but the *R* value of the deposition at Tsukuba decreases.

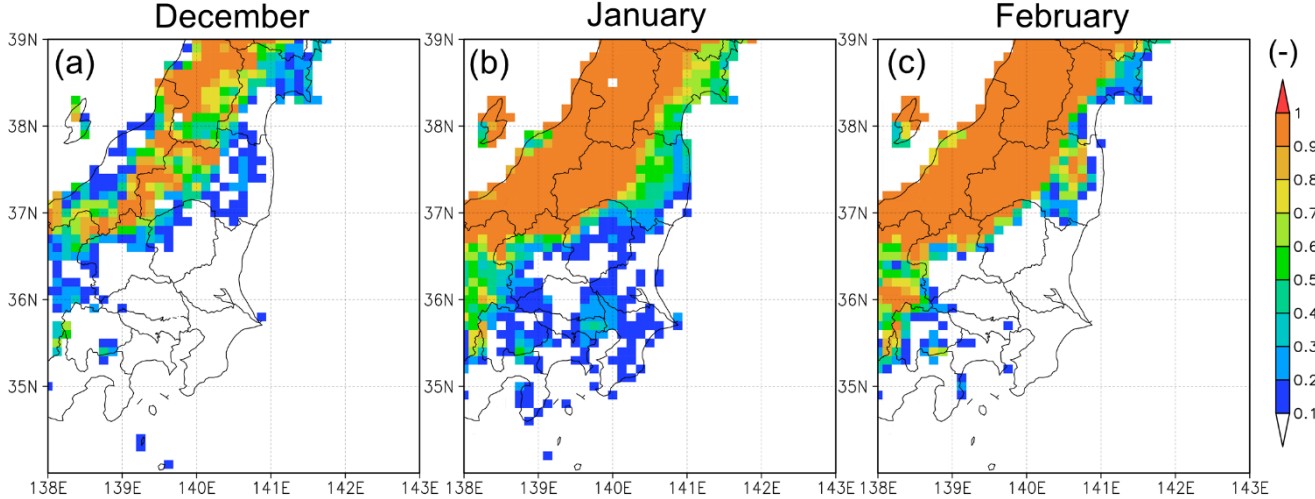

**Figure 9:** Monthly mean MODIS snow cover fractions interpolated to the model grids.

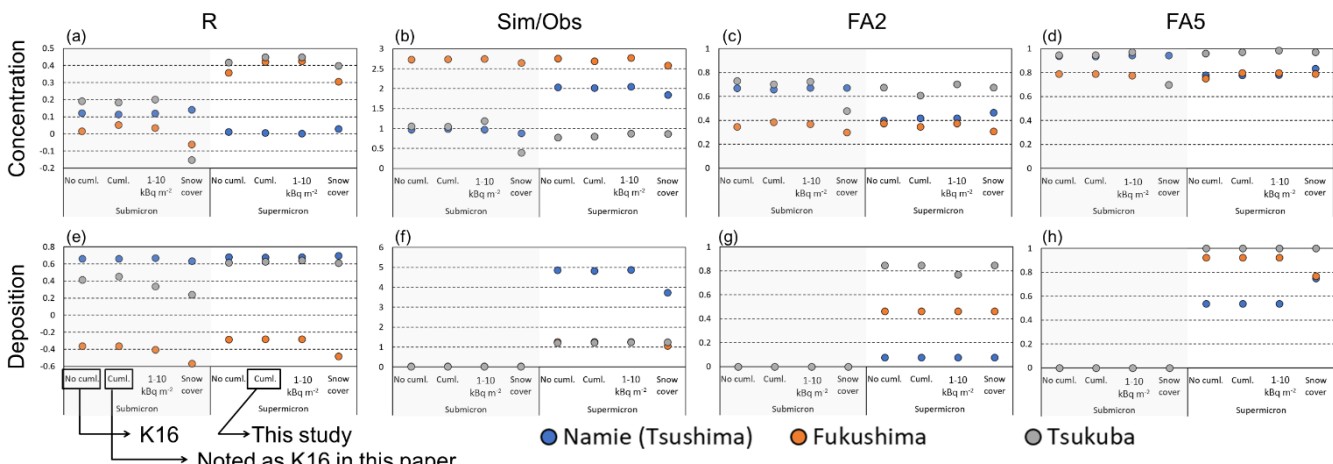

**Figure 10:** Summary of statistical measures between the observations and the simulations (dust plus forest) for various sensitivity tests, such as (from left to right) the correlation coefficient (*R*), simulation-to-observation median ratio (*Sim/Obs*), numerical fraction of data within a factor of two (*FA2*), and numerical fraction of data within a factor of five (*FA5*) for (top) the surface concentrations and (bottom) depositions at (blue) Namie (Tsushima), (orange) Fuklushima, and (gray) Tsukuba. The four sensitivity tests are conducted for both the submicron and supermicron cases and consider no cumulus convection

parameterization [No cuml.], the addition of the cumulus convection parameterization [Cuml.], cumulus convection plus emissions from grids in which the initial deposition amounts are 1 – 10 kBq m⁻² [1–10 kBq m⁻²], and cumulus convection plus emissions from the 1 – 10-kBq m⁻² areas plus suppressed emissions in the presence of snow coverage [Snow cover]. Thus, "Submicron of [No cuml.]" is the simulation setting of K16, "Submicron of [Cuml.]" is the setting denoted as K16 in this paper, and "Supermicron of [Cuml.]" is the setting of this study.





**Figure 11:** (Left) Temporal variations in the surface concentrations of (black) observed [137]Cs and simulated [137]Cs associated with (red) dust aerosols (denoted as [1–10 kBq m[-2]] in Fig. 10), (pink) dust aerosols with emissions suppressed by surface snow coverage (denoted as [Snow cover] in Fig. 10), and (lime) forest aerosols (denoted as "1–10 kBq m[-2]" in Fig. 10), assuming supermicron sizes of [137]Cs-bearing particles at (top to bottom) Namie (Tsushima), Fukushima, and Tsukuba (mBq m[-3]). The sampling intervals are used for the observations, but weekly mean values are depicted for the simulations. (Right) Same as the left panels but for the monthly cumulative deposition amounts (Bq m[-2]).

Among the sensitivity tests, implementation of the cumulus convection parameterization [Cuml.] and the inclusion of less-contaminated areas [1–10 kBq m[-2]] do not cause any substantial differences in the performances of simulating the concentrations and depositions, but the difference induced by including the snow cover effect [Snow cover] is significant. The supermicron simulations indicate that including snow cover improves the performance at Namie (Tsushima) (indicated by the *Sim/Obs* and *FA5* values of the deposition) but deteriorates the performance at Fukushima (as indicated by the *R* value of the concentrations and the *FA5* value of the deposition). Fig. 11 compares the simulated dust between these two settings, [1–10 kBq m[-2]] and [Snow cover], to analyze the concentrations and depositions of the supermicron case at the three sites. Apparently,





[Snow cover] improves both the concentration and deposition simulations at Namie (Tsushima) in January, February, and
December, but deteriorates both the concentration and deposition simulations at Fukushima. This result is consistent with the
fact that no decontamination work occurred in the DRZ around Namie (Tsushima) in 2013, so snow cover suppressed
resuspension from the bare soil. On the other hand, people lived in Fukushima city and the surrounding municipalities, so
snow removal operations (deicing agents and snowblowers) are applied after each snowfall. In fact, substantial amounts of
road salts are observed in road-side $PM_{10}$ measurements in Nordic countries in winter (Denby et al., 2016), indicating the
presence of road dust emissions after snow removal operations. In Fukushima city in 2013, most public facilities and
agricultural fields were already decontaminated, but the achievement ratios of decontamination on roads and forests were
lower than 10% (Watanabe et al., 2021). Thus, snowfall did not suppress the dust emissions around Fukushima city, which
may be the reason why [Snow cover] deteriorated the model performance at the Fukushima site. In addition, as previously
discussed, we hypothesize that the deposition amount in January 2013 at Fukushima was much higher than that at Namie due
to the opposing impacts of snow cover on dust emissions over the two different locations: the suppression around the Namie
site and the production of very large road dust particles around the Fukushima site.

## 4 Conclusions

The regional budget of resuspended $^{137}Cs$ originating from the Fukushima nuclear accident assessed by Kajino et al. (2016)
(K16) for 2013 is reassessed in this study. K16 assumed resuspension aerosol sizes similar to those of primary emissions (the
direct emissions from the F1NPP associated with the accident), which are submicron-sized. However, Watanabe et al. (2021)
determined that the deposition amounts simulated by K16 were significantly underestimated. Based on recent cumulative
knowledge, major resuspension aerosols could be supermicron-sized, such as soil dust (Ishizuka et al., 2017; Kinase et al.,
2018) and bioaerosols (Kinase et al., 2018; Igarashi et al., 2019b; Kita et al., 2020; Minami et al., 2020; Igarashi, 2021). Lower
possibilities of submicron particle involvement, such as that resulting from forest fires (Kinase et al., 2018) and epicuticular
wax (Nakagawa et al., 2018), have been reported. Thus, the regional budget considering supermicron aerosols is significantly
different from that considering submicron aerosols: faster supermicron deposition rates necessitated higher emission fluxes to
sustain the simulated surface concentrations at the observed levels.

To evaluate the simulations, measured concentration and deposition data obtained at three stations, Namie (Tsushima),
Fukushima, and Tsukuba, are used. In this study, the resuspension source area is defined as areas in which the initial deposition
amounts exceed 300 kBq m$^{-2}$. The Namie (Tsushima) site (2300 kBq m$^{-2}$) is in the resuspension source area and is surrounded
by mountainous forests in the Abukuma Highlands. The Fukushima site (190 kBq m$^{-2}$) is characterized as an urban/rural region
located outside but nearby the source area. The Tsukuba site (21 kBq m$^{-2}$) is characterized as a downwind region. A source-
receptor relationship analysis is performed, and resuspension ratios and redistribution amounts are derived. The effects of snow



cover on resuspension and the contributions of resuspension to the actual decreasing trends in the ambient gamma dose rates
are discussed.

The major findings in the context of contrasting the two different particle sizes are summarized as follows.

· Regarding the submicron particles, the surface concentrations of $^{137}$Cs at Namie (Tsushima) in winter are quantitatively
explained by multiplying the dust emission scheme of Ishizuka et al. (2017) by five, but these values are significantly
underestimated in the summer. Additional forest emissions with a factor of $10^{-7}$ $h^{-1}$ explain the enhancement of the
observed $^{137}$Cs surface concentrations in summer at Namie (Tsushima). However, this effect causes opposite seasonal
variations at the Fukushima site: the simulated concentrations are high in summer, but the observations are low in summer.
In addition, this factor causes deposition underestimations by two orders of magnitude at all sites, Namie (Tsushima),
Fukushima, and Tsukuba. The annual mean source contributions for the concentrations are 90%, 16%, and 21%, and those
for the depositions are 95%, 13%, and 20% for Namie (Tsushima), Fukushima, and Tsukuba, respectively. The total areal
annual resuspension of $^{137}$Cs is 1.28 TBq, which is equivalent to only 0.048% of the initial deposition in March 2011, i.e.,
2.68 TBq. The decreasing trend of the observed gamma dose rate in Fukushima Prefecture was $1.0–7.9×10^{-4}$ $d^{-1}$ in 2013
after the radioactive decay of $^{134}$Cs and $^{137}$Cs was excluded. The decreasing trend is due to decontamination and natural
decay, such as that occurring due to land surface processes. The resuspension rate through the atmosphere is 0.048% $y^{-1}$
($1.3×10^{-6}$ $d^{-1}$), which is negligibly small compared to the decreasing trend. Together with the discharge rate through rivers
estimated as 0.02–0.3% $y^{-1}$ (Iwagami et al., 2017), K16 concluded that ground-surface $^{137}$Cs stays or circulates within
local terrestrial ecosystems and is hardly discharged through the atmosphere or rivers.

· Regarding the supermicron particles, by using the climatological deposition velocity analysis proposed by Watanabe et al.
(2021), the dry and wet deposition parameters are successfully constrained by the concentrations and depositions measured
at the three sites. The constrained dry and wet scavenging rates of supermicrons are 100 times and 10 times those of
submicrons, respectively, resulting in the emission fluxes of both dust and forest aerosols to be enhanced twenty-fold.
Compared to the submicron case, the source contributions of supermicrons are higher in the source areas and lower in the
receptor regions. The annual mean source contributions for the concentrations are 94%, 9.1%, and 5.4%, and those for the
depositions are 99.5%, 2.2%, and 1.0% at Namie (Tsushima), Fukushima, and Tsukuba, respectively. The areal total
annual resuspension of $^{137}$Cs is 25.7 TBq, which is equivalent to 0.96% of the initial deposition. Due to the rapid deposition
rates, the annual redeposition amount is also large, at 10.6 TBq; thus, approximately 40% of emissions are redistributed
over East Japan. However, the traveling distance should not be large because the source contributions of the depositions
at Fukushima and Tsukuba are only 2.2% and 1.0%, respectively. The resuspension rate through the atmosphere is 0.96%
$y^{-1}$ ($2.6×10^{-5}$ $d^{-1}$), which may not be negligibly small, as it can account for 1–10% of the decreasing rate due to
decontamination and natural decay except radioactive decay ($1.0–7.9×10^{-4}$ $d^{-1}$). The areas with positive redistribution





amounts (enhanced deposition due to resuspension) of 1–10 Bq m$^{-2}$ are much larger for the supermicron case than those for the submicron case, and the maximum values are up to 100 Bq m$^{-2}$, especially over the ocean close to the coast of Fukushima Prefecture.

From the current analysis, it is likely that snow cover in winter (January, February, and December) suppressed the dust emissions in the source areas around the Namie (Tsushima) site but did not suppress emissions around the Fukushima site. This is because Namie (Tsushima) is located in the DRZ and human activities in this region were very limited in 2013, whereas snow removal operations involving deicing agents and snow blowers were performed in Fukushima city and the surrounding municipalities at this time. In addition, heavy traffic on the major roads close to the Fukushima site (< 1 km) may produce substantial numbers of superlarge road dust (or road salt) particles (~100 μm, which can travel only 1 km) from wet
and muddy surfaces, which may cause exceptionally large deposition amounts in Fukushima in January. The completion of decontamination in 2013 was lower than 10% for roads and forests in Fukushima city.

More than ten years have passed since the accident but the issues to be resolved in the future are still the same as those listed in K16. The current study represents an order estimation of the regional budget for only one year using a simple model and schemes. In addition to the utilized model and schemes, the current horizontal grid resolution is too coarse to reflect
the heterogeneous distributions of various land use types. Soil dust and road dust emissions are relatively well-formulated, but bioaerosols are not. Substantial efforts have been made to understand the emission mechanisms and quantifications of bioaerosol emission fluxes (Igarashi et al., 2019b; Kita et al., 2020; Minami et al., 2020), but it is still difficult to establish a set of formulas that is applicable for various vegetation surfaces. Our hypothesis of the existence of superlarge particles is not proven at all. The decreasing trends in atmospheric $^{137}$Cs differ between the periods before and after approximately 2015
(Watanabe et al., 2021), but the reason for this distinction is not clear. A long-term (i.e., 10-year) assessment using long-term measurements and numerical simulations is required. The quantification and formulation of size-resolved $^{137}$Cs emission fluxes from various sources should directly connect to the comprehensive understanding of the regional budget of resuspended $^{137}$Cs.

**Data and code availability**

The observation and simulation data used for the figures and source codes of the LM are available at https://mri-2.mri-
jma.go.jp/owncloud/s/Cr6nS3iJXPTZLf7 (last access: 20 August 2021).

**Author contribution**

MK developed the numerical model with RH and performed the numerical simulations with MI and KI. AW conducted the measurements at Fukushima University, and KK and TK conducted the concentration measurements at Namie (Tsushima). YS, HH, NA, MH, and ST supported the Namie (Tsushima) measurements and data analysis of the deposition measurements.



YZ, YI, and TK conducted the measurements at Tsukuba. KK and AS supported the data analysis and figure generation. MK designed the manuscript structure and completed the draft together with all authors.

**Competing interests**

The authors declare that they have no conflicts of interest.

**Acknowledgments**

This work was supported by Grants-in-Aid for Scientific Research (KAKENHI) with Grant Numbers JP19H01155 and JP16KK0018. This work was also supported by the Japanese Radioactivity Survey from the Nuclear Regulation Authority (NRA), Japan, the Environmental Research and Technology Development Fund (JPMEERF20215003) of the Environmental Restoration and Conservation Agency of Japan (ERCA), and Arctic Challenge for Sustainability II (ArCS II) with Grant Number JPMXD1420318865 from the Ministry of Education, Culture, Sports, Science, and Technology Japan (MEXT).




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
