# Peer review of "Reassessment of the radiocesium resuspension flux from contaminated ground surfaces in East Japan"

_Atmospheric Chemistry and Physics, 2021_

## Author Comment (AC2)

**Dear anonymous referee #2,**

We are very grateful for your detailed review and constructive comments and your time for RC2. Thanks to your review, our manuscript has been substantially improved, especially for readability (and I am very sorry for many typos which took your time). We have considered all your comments in the revised manuscript.

Point-by-point responses to your comments are written in blue in this letter.

With best regards, Mizuo Kajino

[1] The manuscript touches an important aspect of atmospheric transport modelling, namely the assumptions made regarding aerosol size distributions. Often size distributions employed are simplified or even inappropriate. I think this work is an important contribution towards better understanding the issue of aerosol size-distributions. Overall, the paper is well written and I can recommend its publication in ACP. There are, however, some things which need clarification or correction.

[1] Thank you for your evaluation.

**Specific comments:**

[2] Fig. 1: source-receptor analysis presented in Sect. 3.5 -> source-receptor analysis presented in Sect. 3.2

[2] I changed it, thank you.

Section 2.1 Observation data:

[3] A few words about the sampling devices used for activity concentration and deposition measurements would be good.

[3] Yes, they are necessary to mention. Sampling devices such as "high-volume air samples", "plastic tray", and "germanium detectors" are mentioned for the explanations of each observation site. Please also refer to our reply to [1] of RC1.

[4] I. 130: The Namie (Tsuhima) site is approximately 30 km northwest of the F1NPP, is -> The Namie (Tsushima) site [skip "is"] approximately 30 km northwest of the F1NPP [skip comma] is

[4] I deleted it.

[5] I. 151/152: (only living areas; the removal of shrub and litter layers within 20 meters from the forest edges) Please rephrase. Not fully clear what is meant.

[5] We meant that "within 20 meters" are defined as "living areas". We removed the term "living areas" and rephrased as follows:

(only removal of shrub and litter layers in areas within 20 meters from the forest edges)

Section: 2.2 Numerical simulations:

[6] The first paragraph of this section is not well written.

[6] Changes according to your comments are described line by line.

- Errors between the original 1-D Eulerian model and the 1-D Lagrangian model developed and implemented to LM: I guess you rather mean "deviations" instead of "errors". 1-D Lagrangian model was implemented to LM? Reads strange. You need to mention that a 3D model was employed. Evidently it is built on previous 1D models.
- I changed the sentences to "The deviations between the original 1-D Eulerian convection model and the 1-D Lagrangian convection model developed in the current study are summarized in Supplement 2 for evaluation of the convection scheme implemented in 3-D LM"
- What was the number of particles employed? What is the temporal resolution of input and output? Actually I would recommend a table with the run set-up.
- This information was not repeated here as they were written in our previous study. However they are important so we made a new Table 1.
- from 138–140°E -> from 138–143°E
- Thank you. I changed it.
- Grid Point Value-Mesoscale Model (GPV-MSM): Reference is missing.
- The web page https://www.jma.go.jp/jma/en/Activities/nwp.html, was inserted in the main text
- The GPV-MSM provides data on three hourly meteorological variables on the surface and at vertical layers from 1000 hPa to 100 hPa. -> The GPV-MSM provides data for meteorological variables with 3-hourly resolution on the surface

and at vertical layers from 1000 hPa to 100 hPa. How many layers are there?

- I changed it. 16 layers.
- I have never heard the term "hydrometeor concentrations". Anyway: The "fundamental variables" need in any case to include precipitation rate P for wet deposition.
- Probably, mixing ratios of hydrometeors should be better. Thank you for your suggestion on P.
- to simulate the locations and masses (or radioactivities) of Lagrangian particles -> to simulate the locations and masses (or activity) carried by Lagrangian particles.
- I changed it.
- The simulation period is from December 1, 2012, to January 1, 2013, and the analysis period is the full year of 2013, from January 1, 2013, to January 1, 2014. Do you mean in fact a spin-up period? Not clear.
- It was typo: The simulation period is from December 1, 2012 to January 1, 2014. I corrected it and changed a sentence a little bit.

[7] I. 213: If the unit of FM is kg m-1 s-1 then the units on the left and right side of eq. 3 do not match. "as a function of the friction velocity": Please add dependency to eq. 3.
[7] Sorry for the typo. It must be kg/m2/s. I also added the equation and reference (Loosemore and Hunt, 2002).

[8] Eq. 4: No reference is provided.[8] K16 is the reference for Eq. 4.

[9] I. 227: derived from Advanced Very High Resolution Radiometer (AVHRR) normalized difference vegetation index (NDVI) data: Reference is missing.

[9] https://www.usgs.gov/core-science-systems/eros/phenology/science/ndvi-avhrr was inserted.

[10] I. 235: additionally consider areas with deposition amounts of 1–10 kBq m-2 How can deposition amounts be quantified if below the detection limit?

[10] The value became below 10 kBq m-2 after horizontal interpolation. We modified the paragraph as follows:

"For both the dust and forest aerosol cases, only emissions from the grids in which the mean initial deposition amounts exceed 10 kBq m-2, the detection limit of the airborne

measurements, are considered (NRA, 2012). However, horizontal interpolation from the original grids to the LM grids can convert values slightly above the detection limit to less than 10 kBq m-2. Thus, a sensitivity test is performed to additionally consider areas with deposition amounts of 1–10 kBq m-2 as emission sources, as is presented in Sect. 3.4."

We also changed the relevant sentence in the first paragraph of Sect. 3.4 in the same manner.

[11] I. 237: NRA, 2014: Reference does not exist. NRA, 2012?[11] NRA, 2014 was missing and inserted in the reference list.

[12] I. 263/264: the typical deposition rates of major carrier aerosols (submicron aerosols) are much smaller than the resuspension rates (supermicron aerosols). -> the typical deposition rates of major carrier aerosols (submicron aerosols) are much smaller than those of the major carriers in this study (supermicron aerosols). ?

[12] Yes. We rephrased it to "The reason for this underestimation of the deposition amounts is obvious; the typical deposition rates of major carrier aerosols (submicron aerosols) of K16 are much smaller than those of soil dust and bioaerosols (supermicron aerosols)."

[13] I. 278-280: This analysis is novel because emission flux adjustments ( $C_{const}$  in Eq. 3 and  $r_{const}$  in Eq. 4) do not change the slope of the regressions, so the deposition parameters can be adjusted independently from the emission flux adjustment. Please explain. Not obvious to me. Why should the two parameterizations be related at all? [13] In this study, we needed to adjust both emission flux and deposition flux. If both emission and deposition affect the slope, the slope adjustment must be tough. Because emission does not affect the slope, we can adjust deposition first to fix the slope and then adjust the emission. We made a new paragraph to elaborate it as follows:

"This analysis is novel because emission flux adjustments do not change the slope of the regressions, so the deposition parameters can be adjusted independently from the emission flux adjustment. In this study, we need to constrain both emission and deposition parameters. If emission and deposition parameters both affect the slope, the adjustment should be tough. However, in this analysis we constrain the deposition parameters first to fix the slope (or the intercept of y-axis) with whatever emission parameters, and then constrain the emission parameters using observed concentration and deposition values as presented in the next subsection (2.3.2)."

[14] I. 289: are efficiently collected deposition samplers, as the traveling distance is approximately 1 km -> are efficiently collected by deposition samplers. I think the last part of the sentence can be skipped. It looks lost to me.

[14] Thank you. The two sentences are separated as follows:

"are efficiently collected deposition samplers. The traveling distance of such superlarge particles is estimated as approximately 1 km (e.g., Kajino et al., 2012; Kajino et al., 2021)."

[15] I. 332: Thus, for simplicity, we multiplied both fluxes used in K16 by 20: Based on which considerations?

[15] We changed the sentence as follows:

"Thus, we simply multiplied both fluxes used in K16 by 20 so that number fractions of data within a factor of five exceeded 0.5 for both concentrations and depositions at the three sites:  $C_{const}$  was 100 and  $r_{const}$  was 2×10-6 h-1."

Section: 3.1 Seasonality and quantity of surface air concentrations and depositions

[16] I. 348: around the site (over the plain), the 137Cs of dust aerosols -> around the site (of the plain), the 137Cs deposition of dust aerosols
[16] We changed it.

[17] I. 351: (Sim/Obs = 4.8) -> (Sim/Obs = 4.9) Value in the text and that in the figure should agree.

[17] Thank you. We changed it.

[18] I. 357: there are certain areas of snow coverage -> there are certain periods of snow coverage[18] We changed it.

[19] I. 390: the simulations are high in summer due to forest aerosols -> the simulations are too high in summer due to forest aerosols for the submicron case[19] We changed it.

[20] I. 393-395: The observed surface concentrations are high at Fukushima in winter, and the observed short-term peaks correspond to the simulated dust aerosols, indicating that the emission of resuspended 137Cs at Fukushima in winter is driven by wind. Not only in winter. There are also similar peaks in the observations in late spring/early summer.

[20] Thank you. We have added the following additional sentences:

"The simulated forest aerosols are approximately one order of magnitude smaller than the dust aerosols in winter. The simulated dust peaks also correspond to the observed peaks in later spring and early summer. The simulated contribution of forest aerosols is as great as dust aerosols during the season, but there may also be an association of wind-induced dust aerosols."

[21] I. 398/399: Nevertheless, Sim/Obs is not very low (0.80), and the R value obtained for supermicrons is improved from the submicron case -> The R value obtained for supermicrons is improved compared to the submicron case. Skip the reference to Sim/Obs – Sim/Obs with a value of 1.05 is even better for the submicron case. So Sim/Obs of the supermicron case is in fact not in favor of assuming larger particles.
[21] Thank you. We changed the sentence accordingly.

[22] I. 406/407: that the numbers presented in these sections are associated with the discrepancies in the simulations described in the current section. Please rephrase. Not at all clear.

[22] The full sentence was modified as follows: "In the following subsections (Sects. 3.2 and 3.3), the source-receptor relationship and annual resuspension ratios are discussed, although the simulation is associated with the discrepancies presented in this section."

[23] I. 424/425: are derived using the seasonal mean concentrations from 300 kBq m-2 areas divided by those from whole areas (i.e., > 10 kBq m-2) -> are derived using the seasonal mean activity concentrations resulting from 300 kBq m-2 areas divided by those from the overall area (i.e., > 10 kBq m-2) [23] We changed it accordingly.

[24] I. 434/435: Even though the surface concentrations of supermicron particles over Fukushima Prefecture are larger than those of submicron particles (Fig. 5b), -> Even

though the surface concentrations of supermicron particles (Fig. 5b) over Fukushima Prefecture are larger than those of submicron particles (Fig. 5a), [24] We changed it accordingly.

[25] I. 436: < 0.01 mBg m-3 -> < 0.01 mBq m-3 Analogous misspelling in I. 542. [25] Thank you. We changed both.

[26] I. 445-447: Even though the seasonal mean wind fields over the ocean close to land are directed toward the land, substantial proportions of 137Cs in forest aerosols are transported toward the ocean in summer Please elaborate what is the mechanism behind this feature.

[26] We changed the sentence as follows:

"The seasonal mean wind fields over the ocean close to land are directed toward the land, indicating that ocean-to-land wind blows are more frequent (and/or stronger) than land-to-ocean wind blows. Thus, less frequent and/or weaker land-to-ocean wind transported substantial proportions of 137Cs in forest aerosols toward the ocean in summer..."

[27] I. 448: the 137Cs transported toward the ocean are transported -> the 137Cs transported toward the ocean is transported
 [27] We changed it.

[28] I. 473/474: the source contributions of submicrons (Fig. 6c) are similar to those for the concentrations (Fig. 6a), but the source contributions of supermicrons (Fig. 6d) are remarkably different. Please elaborate why this is the case. I find it difficult to understand why the contributions from the re-suspension area is that different for concentration and deposition for the super-micron case.

[28] The old explanation

"As presented later in Fig. 7, approximately 40% of supermicron emissions were deposited in the same grid. Thus, the local deposition contributions become much larger than the concentration contributions."

are replaced by the new one as

"Removal rates of submicron particles are small so that they do not affect surface concentrations (D = aC, recall Eq. 5) and thus Figs. 6a and 6c are similar. However, the large removal rates of supermicron particles can substantially alter the surface

concentrations (D = aC(D)). For the supermicron case, both emissions and depositions are substantially enhanced with compared to the submicron case. The enhanced depositions (by almost two orders of magnitude, see Fig. 3) directly alter the source contributions for depositions (Fig. 6d). However, changes in concentrations due to enhanced emissions (set as 20 times) are compensated by enhanced depositions, and thus the source contributions for concentrations of supermicrons (Fig. 6b) are slightly but not drastically different from the submicron cases (Fig. 6a)."

[29] I. 490: (Figs. 7c–7d). -> (Figs. 7a–7d).[29] We changed it.

[30] I. 490/491: resuspension amount was 1.28 TBq (Fig.7a) Please cite Figs 7b to 7d in an equivalent manner.

[30] We modified it accordingly.

[31] I. 512/513: the positive redistribution area (area enhanced deposition due to resuspension) is limited, and the amounts are up to 10 Bq m-2 per year. On the other hand, for the supermicron case (Fig. 8a), -> the positive redistribution area (area with enhanced deposition due to resuspension) is limited, and the amounts are up to 10 Bq m-2 per year. On the other hand, for the supermicron case (Fig. 8d), [31] We changed it.

Section: 3.4 Sensitivities: This section contains some weaknesses.

[32] I. 545/546: While this parameterization is complicated, the differences between [Cuml.] and [No cuml.] are exceedingly small. Can be removed. The statement reappears in a similar manner in the 3rd paragraph.

[32] We removed the sentence.

[33] I. 553/554: and over all prefectures numbered in Fig. 1b except the western part (Ou Mountains) of Fukushima Prefecture) Please remove. First, it is Fig. 1a where prefectures are numbered and second there are prefectures (e.g., #4) with elevations > 1000 m.

[33] Thank you. We removed it.

[34] I. 554: including at the Fukushima -> including [skip "at"] the Fukushima

**[34] We removed it.**

[35] I. 558-561: Should be removed completely. The discussion of submicron vs. supermicron is redundant (see section 3.1). Also, please avoid general statements which are not supported by the figures and only use the term "significant" if significance testing was performed.

[35] We removed the whole sentence. Also, thank you for the use of "significant". We rephrased all "significantly" and "significant" in the manuscript to other phrases such as "substantial", "remarkable", and "evident".

[36] I. 561/562: Including the cumulus parameterization was successful in the sense that it did not cause any significant deterioration in the statistical scores. Strange sentence. A cumulus parameterization can be considered successful if it improves results but not when it does not deteriorate results!

[36] We should have said that including cumulus was not failed as it did not cause any deterioration. We rephrased the sentence as follows: "Including the cumulus parameterization did not show any substantial changes in the statistical scores."

[37] I. 562-566: The supermicron simulations indicate slight improvements due to cumulus convection, such as increased R values of the concentrations at Fukushima and Tsukuba, but the FA2 values of the concentrations at Fukushima and Tsukuba are slightly decreased. The impact of including a 1–10 kBq m-2 area would be larger at Tsukuba, which is surrounded by less contaminated regions. The supermicron simulations indicate a slight improvement in the R value of the concentrations at Tsukuba, but the R value of the deposition at Tsukuba decreases. Please remove these lines. There is no use in discussing the score differences detail, if these differences are deemed not substantial right afterward. And looking at Fig. 10 the statement about the change of R for Tsukuba when including deposition areas with 1-10 kBq m-2 is simply not true.

[37] We removed the whole sentences.

[38] Fig.10: I would remove those parts referring to the submicron results as this topic is already covered in section 3.1. This would help to make the subplots larger and labels easier to read (currently quite difficult due to font size). Caption: Thus, "Submicron of [No cuml.]" is the simulation setting of K16, "Submicron of [Cuml.]" is the setting denoted as K16 in this paper, and "Supermicron of [Cuml.]" is the setting of this study. Please

remove entirely. Quite confusing. Rather explain in the first paragraph of the section that K16 simulations were redone using cumulus parameterization.

[38] Thank you. We removed the submicron panels from Fig.10 and the confusing description from its caption.

[39] Fig. 11: Add "(without snow cover)" in the legend for the red line.[39] We added it.

[40] I. 589: significant Again, please use this term only in conjunction with a significance test.

[40] We changed it to "substantial".

[41] I. 603-606: In addition, as previously... Please remove. Redundant to what is said in the same paragraph above.

[41] We removed the whole sentence.

In addition, authors could considers making the following changes: Thank you again for your detailed investigation.

[42] I. 51: which is not included -> which are not included

[43] I. 73: In addition to spatial observations -> In addition to spatiotemporal observations

[44] I. 81: Vertical measurements obtained on mountains -> Altitude-dependent measurements obtained on mountains

[42-44] We changed it accordingly.

[45] I. 83: in other models: Which other models?[45] We changed from "other" to "almost all".

[46] I. 105/106: In fact, number of pollen particles was 1/10 of number of bacteria -> In fact, the number of pollen particles was 1/10 of the number of bacteria

[47] I. 113: the major carriers of 137Cs -> the major carriers of re-suspended 137Cs

[48] I. 136: The concentration measurements are conducted -> The concentration measurements were conducted

[49] I. 137/138: the deposition measurements are made by Fukushima Prefecture at Tsushima Screening Center -> the deposition measurements were made by Fukushima

Prefecture at the Tsushima Screening Center

[50] I. 144: The concentration and deposition measurements are -> The concentration and deposition measurements were

[51] I. 155/156: The concentrations and deposition amounts are measured -> The concentrations and deposition amounts were measured

[52] I. 200/201: are the orders of these submicron particles -> are valid for these submicron particles

[53] I. 203: improved from those used in K16 -> improved compared to those used in K16

[54] I. 207/208: K16 simulated the contributions from these additional emissions as being two to three orders of magnitude smaller than the observed surface activity concentrations, -> K16 simulated the contributions from these additional emissions and resulting activity concentrations were two to three orders of magnitude smaller than the observed surface activity concentrations,

[55] I. 232: only emissions from the grids -> only emissions from the grid boxes

[56] I. 245: or Tsukuba sites, and -> or Tsukuba sites, but

[46-56] We changed all accordingly.

[57] I. 261: at Fukushima sites -> at Fukushima city sites

[57] Sorry for the confusion. It is the Fukushima site, same as the one defined in this paper. It was changed to "at the Fukushima site".

[58] I. 273: which is on the dimension -> which is of the dimension

[59] I. 300: deposition rates obtained in this study are much faster -> deposition rates obtained in this study are much higher

[60] I. 353: we regard this application -> we regard a uniform application

[61] I. 393: due to forest aerosols is less dominant -> due to forest aerosols is less dominant for the supermicron case

[62] I. 398: compared to the observations in winter -> compared to the observations in winter in both cases

[63] I. 401: 100 -> 1

[64] I. 421: is defined as grids in which the grid-mean initial deposition -> is defined as domain in which the grid box mean initial deposition

[65] I. 450/451: over Ibaraki and Miyagi exceeded 30% at a limited number of grids, but the mean concentrations were much lower (Fig. 5d) than those in the submicron case (Fig. 5c) over the prefectures -> over Ibaraki and Miyagi exceed 30% at a limited

number of grid boxes, but the mean concentrations are much lower (Fig. 5d) than those in the submicron case (Fig. 5c) over these prefectures

[66] I. 537: from the grids in which the grid-mean -> from the grid boxes in which the grid box mean. I would also recommend replacing "grids" with "grid boxes" in the subsequent lines.

[67] I. 543-545: Thus, the submicron case of [No cuml.] was used in the study of K16, and the supermicron case of [Cuml.] is used as the simulation in this study. Note that the submicron simulations shown in the current study are [Cuml.].-> Thus, the submicron case with [No cuml.] was used in the study of K16, and the supermicron case with [Cuml.] is used as the reference simulation in this study. Note that the submicron simulations shown in the current study are also with [Cuml.].

[68] I. 555/556: Some areas over the Kanto Plain are covered with snow in January. Extensive snow cover is also observed in the Abukuma Highlands, including in Namie (Tsushima), in February. -> Some areas over the Kanto Plain are also covered with snow in January. Extensive snow cover is in addition observed in the Abukuma Highlands, including [skip "in"] Namie (Tsushima), in February.

[69] I. 576: emissions from grids in which the initial deposition -> emissions from grid boxes where the initial deposition

[70] I. 597: people lived in Fukushima -> people live in Fukushima

[71] I. 603: why [Snow cover] deteriorated -> why [Snow cover] deteriorates

[72] I.616: faster supermicron deposition rates necessitated higher emission fluxes -> faster supermicron deposition rates necessitate higher emission fluxes

[73] I. 619: the resuspension source area is defined as areas in which -> the resuspension source area is defined as area where

[74] I. 629: Additional forest emissions with a factor of  $10^{-7}$  h-1 -> Additional forest emissions with applying a constant factor of  $10^{-7}$  h-1

[75] I. 633: The annual mean source contributions for the concentrations -> The annual mean source contributions of the re-suspension source area for the concentrations

[76] I. 648: 9.1%, and 5.4% Floating point vs. integer notation in section 3.1

[77] I. 657: the maximum values are up to 100 Bq m-2, -> the maximum values are up to 100 Bq m-2 for the supermicron case,

[78] I. 665: which may cause exceptionally -> which may in turn cause exceptionally[79] I. 673/674: is not proven at all. -> is not proven yet. I would not be that strict.

[58-79] We changed all accordingly.